# WeGeFT: Weight-Generative Fine-Tuning for Multi-Faceted Efficient Adaptation of Large Models

**Chinmay Savadikar** [1]  **Xi Song** [2]  **Tianfu Wu** [1]
Code:  https://savadikarc.github.io/wegeft

## Abstract

Fine-tuning large pretrained Transformer models can focus on either introducing a small number of new learnable parameters (parameter efficiency) or editing representations of a small number of tokens using lightweight modules (representation efficiency). While the pioneering method LoRA (Low-Rank Adaptation) inherently balances parameter, compute, and memory efficiency, many subsequent variants trade off compute and memory efficiency and/or performance to further reduce fine-tuning parameters. To address this limitation and unify parameter-efficient and representation-efficient fine-tuning, we propose Weight-Generative Fine-Tuning (WeGeFT, pronounced *wee-gift*), a novel approach that **learns to generate fine-tuning weights directly from the pretrained weights**. WeGeFT employs a simple low-rank formulation consisting of two linear layers, either shared across multiple layers of the pretrained model or individually learned for different layers. This design achieves multi-faceted efficiency in parameters, representations, compute, and memory, while maintaining or exceeding the performance of LoRA and its variants. Extensive experiments on commonsense reasoning, arithmetic reasoning, instruction following, code generation, and visual recognition verify the effectiveness of our proposed WeGeFT.

## 1. Introduction

Fine-tuning pretrained deep neural networks (DNNs) as feature backbones for downstream tasks has been an important and challenging research topic. In recent years, large feature backbones with open weights, such as LLaMA (Touvron et al., 2023a;b; AI@Meta, 2024), have become ubiquitous. Training such models from scratch is infeasible with limited resources, and fine-tuning them entirely can also be prohibitively costly. This raises two critical questions: (i) which parts of a pretrained model should be fine-tuned (often treated as a hyperparameter), and (ii) how those parts should be fine-tuned. In this paper, we focus on the latter question by leveraging module/layer selection strategies widely adopted in prior art.

Low-Rank Adaptation (LoRA) (Hu et al., 2022) is a pioneering and widely adopted approach that achieves built-in efficiency in parameters, compute, and memory. LoRA learns fine-tuning weight residuals in low-rank forms for pretrained weights on a layer-specific basis (see the left of Fig. 1). Thanks to its strong applicability and promising performance, many follow-up works have emerged, such as DoRA (Liu et al., 2024) and VeRA (Kopiczko et al., 2023). However, these variants often sacrifice compute efficiency (training wall time) and/or GPU memory efficiency to achieve reductions in learnable parameters or performance gains on certain downstream tasks. As we demonstrate in experiments, DoRA, while matching or slightly surpassing LoRA's performance, increases training wall time by more than 5x and consumes around 3GB more GPU memory. On the other hand, VeRA, though significantly reducing the number of learnable parameters, performs much worse than LoRA while drastically increasing training wall time (by more than 20x) and consuming similar additional GPU memory. **These trade-offs motivate us to seek a formulation that can significantly reduce the number of learnable parameters, achieve superior or on-par performance compared with LoRA, and retain its efficiency in compute and memory.**

Towards these objectives, the recently proposed ReFT (Wu et al., 2024) introduces a promising framework that focuses on lightweight representation-editing modules instead of learning weight residuals, as LoRA does. ReFT is inspired by causal intervention mechanisms (Geiger et al., 2024) and operates in a layer-specific manner. While ReFT methods

[1]Department of Electrical and Computer Engineering, North Carolina State University, Raleigh, USA [2]An Independent Researcher. Correspondence to: Chinmay Savadikar <csavadi@ncsu.edu>, Tianfu Wu <twu19@ncsu.edu>.

*Proceedings of the $42^{nd}$ International Conference on Machine Learning*, Vancouver, Canada. PMLR 267, 2025. Copyright 2025 by the author(s).

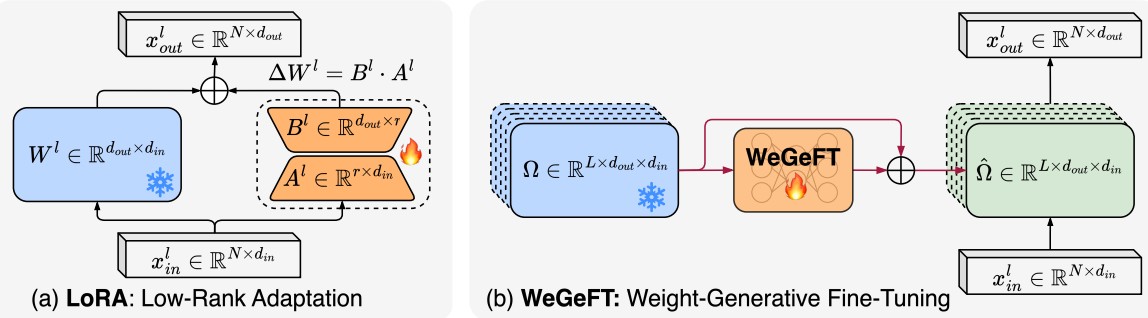

Figure 1: Comparisons between (a) LoRA (Hu et al., 2022) and (b) our proposed WeGeFT.

reduce learnable parameters and retain compute and memory efficiency comparable to LoRA, they often show inferior performance, as confirmed in our experiments. Additionally, selecting where to intervene within the model to achieve strong downstream task performance is non-trivial. For instance, DiReFT, one of the two ReFT formulations, can be interpreted as applying LoRA directly to hidden representations at specific intervention points. **This motivates us to seek a unified perspective between parameter-efficient and representation-efficient fine-tuning that enables simpler formulations while achieving on-par or better performance.**

In summary, our experiments reveal the limitations of LoRA variants such as DoRA and VeRA, as well as alternative methods like ReFT, highlighting their drawbacks in compute, memory, and performance trade-offs. **Outperforming LoRA while maintaining multi-faceted efficiency in parameters, representations, compute, and memory remains a significant challenge.** In this paper, we propose a novel approach to address this challenge.

To clarify the foundation of our proposal, we first review the formulation of LoRA (Hu et al., 2022). Denote the pretrained weights of a layer $l \in L$ of a Transformer model by $W^l \in \mathbb{R}^{d_{out} \times d_{in}}$, and the fine-tuned weights by $\hat{W}^l \in \mathbb{R}^{d_{out} \times d_{in}}$. LoRA is defined as:

$$\text{LoRA:} \quad \hat{W}^l = W^l + B^l \cdot A^l, \quad (1)$$

where $B^l \in \mathbb{R}^{d_{out} \times r}$ and $A^l \in \mathbb{R}^{r \times d_{in}}$ are two low-rank matrices representing new learnable model parameters introduced during fine-tuning, and $r$ is the rank ($r \ll \min(d_{in}, d_{out})$). Typically, $B^l$ is initialized to zeros, while $A^l$ is randomly initialized.

We rethink LoRA from three key aspects:

- **Pretrained Weight Awareness:** LoRA imposes no constraints on $B^l$ and $A^l$ beyond their low-rank structure, enabling downstream task data to dictate the fine-tuning process. However, the pretrained weights $W^l$ encode "carry-over" knowledge that is expected to be useful for downstream tasks. By making fine-tuning weight residuals aware of the pretrained weights, we hypothesize

that performance can be further enhanced, especially for stronger pretrained models. Therefore, we aim to parameterize fine-tuning weight residuals in a weight-aware manner.

- **Layer-Specific vs. Shared Adaptation:** In LoRA, $B^l$ and $A^l$ are layer-specific, which ensures layer-local adaptation. Recent methods like Tied LoRA (Renduchintala et al., 2024) propose sharing $B$ and $A$ across selected layers, while VB-LoRA (Li et al., 2024) introduces a shared vector bank to compose all low-rank matrices via a differentiable top-$k$ admixture. Although these approaches reduce the number of learnable parameters, our experiments show that they often underperform or significantly degrade performance. For instance, Tied LoRA reduces parameter count but sacrifices expressivity, while VB-LoRA adds unnecessary complexity via top-$k$ modeling. *We advocate for parameter sharing across layers but emphasize that the expressivity of the fine-tuned model should rely on pretrained weights to compensate for the reduction in learnable parameters.* Our results highlight the critical role of pretrained weight awareness in enabling this trade-off.

- **Additive vs. Multiplicative Updates:** LoRA uses additive weight residuals ($B^l \cdot A^l$), which can be limiting in terms of expressivity. Alternatively, multiplicative updates may enable richer, structured transformations. For example, DiReFT (Wu et al., 2024) applies multiplicative updates in the representation space. Let $y_i^l \in \mathbb{R}^{d_{out} \times 1}$ denote the activation output (representation) for the $i$-th token at layer $l$. DiReFT updates it as:

$$\begin{aligned} \text{DiReFT:} \quad \hat{y}_i^l &= y_i^l + B^l \cdot (A^l \cdot y_i^l + b^l), \quad (2) \\ &= (\mathbb{I} + B^l \cdot A^l) \cdot y_i^l + B^l \cdot b^l, \end{aligned}$$

where $B^l \in \mathbb{R}^{d_{out} \times r}$, $A^l \in \mathbb{R}^{r \times d_{out}}$, and $b^l \in \mathbb{R}^{r \times 1}$ are parameters of the representation-editing module. $\mathbb{I}$ is the identity matrix. While DiReFT introduces multiplicative residuals into the representation space, it is coupled with token intervention search (token-selective).

To achieve multi-faceted efficiency across parameters, representations, compute, and memory, we propose **Weight-Generative Fine-Tuning** (WeGeFT, pronounced *wee-gift*),

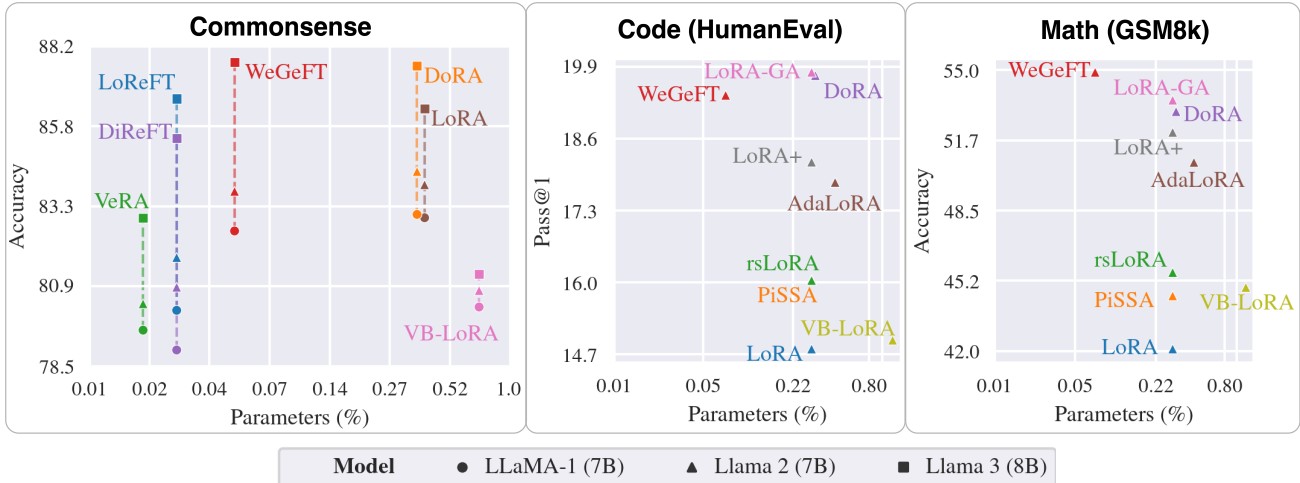

Figure 2: Comparisons of performance vs. trainable parameters between our WeGeFT and baseline methods on three tasks using the Llama model family. Figure 3 shows that WeGeFT maintains the compute and memory efficiency of LoRA, thus achieving very strong multi-faceted efficiency across parameters, representations, compute and memory. See Section 4 for experimental details.

a simple yet effective formulation (see the right of Fig. 1):

$$\textbf{Our WeGeFT:} \quad \hat{W}^l = W^l + W^l \cdot \phi \cdot \psi, \quad (3)$$
$$= W^l \cdot (\mathbb{I} + \phi \cdot \psi),$$

where $\phi \in \mathbb{R}^{d_{in} \times r}$ and $\psi \in \mathbb{R}^{r \times d_{in}}$ are low-rank matrices shared across layers, and $r$ is the rank. See Appendix C for gradient analyses between LoRA and our WeGeFT.

- **Weight-Aware Parameter Sharing:** WeGeFT can be viewed as a weight-aware variant of LoRA, where the layer-specific $B^l$ in LoRA becomes weight-aware ($B^l = W^l \cdot \phi$) and the layer-specific $A^l$ becomes layer-agnostic ($A = \psi$). Compared to Tied LoRA (Renduchintala et al., 2024), WeGeFT retains layer-specific information via $B^l$, preserving performance. Unlike VB-LoRA (Li et al., 2024), WeGeFT avoids the need for a complex shared vector bank and top-$k$ admixture, instead relying on a pair of shared low-rank matrices ($\phi, \psi$) for simplicity and stability. WeGeFT achieves significant improvements in parameter efficiency without sacrificing performance and even enables performance gains when stronger pretrained models (e.g., LLaMA 1 vs. Llama 3) are used.

- **Residual Learning in Weight Space:** WeGeFT extends the residual learning principle of ResNets (He et al., 2016), $x = x + f(x)$, into the weight space. In contrast, DiReFT (Wu et al., 2024) applies this principle in the representation space with causal intervention treatments. WeGeFT, however, eliminates the need to search for specific intervention positions and is token-agnostic, ensuring it retains LoRA's compute and memory efficiency while offering greater flexibility (see Sec. 3.2).

Fig. 2 shows result comparisons on three benchmark datasets, which demonstrate the overall multi-faceted effi-

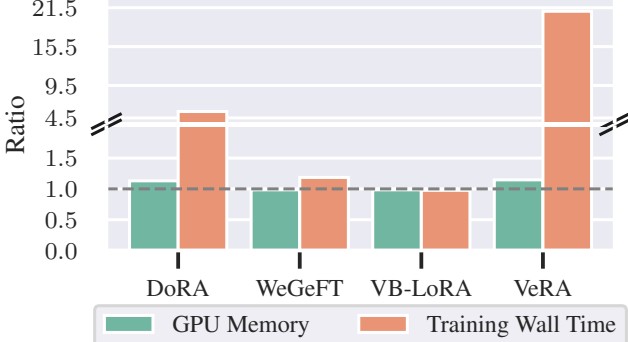

Figure 3: Comparison of the ratio of the GPU Memory (Training Wall Time) for various PEFT methods with the GPU Memory (Training Wall Time) of LoRA. WeGeFT maintains the efficiency of LoRA, as opposed to DoRA and VeRA. Note that while VB-LoRA maintains the memory and compute efficiency, it performs worse than LoRA as seen in Figure 2.

ciency of our proposed WeGeFT. Fig. 3 shows the efficiency comparisons of GPU memory footprint and training wall time.

- **Visual Inspection of WeGeFT in Computer Vision Tasks**: Let $C^l = W^l \cdot \phi$ be the transformation using the first linear layer of WeGeFT for an output projection layer in MHSA (see Sec. 4.5). We show that $C^l \in \mathbb{R}^{d_{out} \times r}$ can be used as a token-clustering head. Using the fine-tuned model, the activation of the output projection layer is, $\hat{y}^l \in \mathbb{R}^{N \times d_{out}}$, where $N$ the number of visual tokens in raster order. We compute $r$ heatmaps for visual token clustering by,

$$H^l_{N \times r} = \hat{y}^l_{N \times d_{out}} \cdot C^l_{d_{out} \times r}, \quad (4)$$

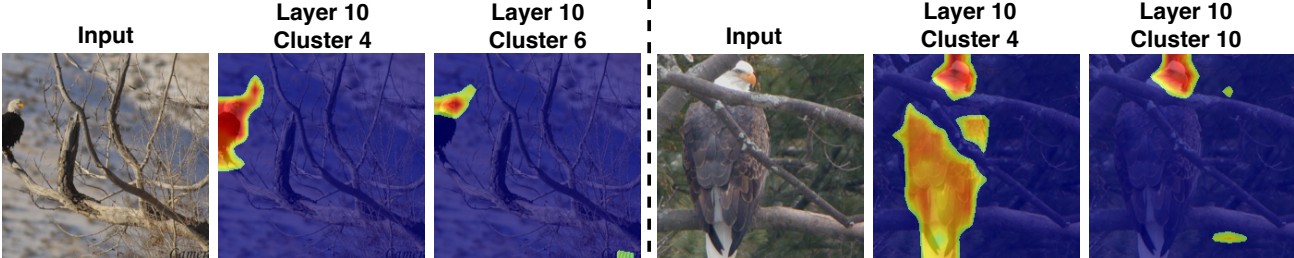

Figure 4: WeGeFT can play the role of a $r$-way token-clustering head that can localize meaningful objects/parts on images. More examples can be found in Figure 5 in the Appendix.

which can highlight semantically meaningful parts of the image. We normalize the $r$ heatmaps to $[0, 1]$ individually and use 0.5 as the threshold in visualizations (Fig. 4).

## 2. Related Work and Our Contributions

**Parameter Efficient Fine-tuning (PEFT).** The goal of PEFT methods is to reduce the computational resources (memory footprint, wall time, etc.) required for fine-tuning large models such as Transformers (Vaswani et al., 2017) and Vision Transformers (ViTs) (Dosovitskiy et al., 2021). Prompt-based methods either append prompts to the input tokens (Lester et al., 2021; Jia et al., 2022), or the intermediate layers (Li & Liang, 2021; Liu et al., 2021; Zhang et al., 2023b). Early work on PEFT used sequential/parallel learnable adapters added after the Multi-Head Self Attention and/or FFN blocks (Houlsby et al., 2019; Bapna & Firat, 2019; Pfeiffer et al., 2021; 2020; Rücklé et al., 2021; Mahabadi et al., 2021a; Chen et al., 2022). LoRA (Hu et al., 2022) and its variants (Zhang et al., 2023a; Dettmers et al., 2023; Lialin et al., 2023; Kopiczko et al., 2023; Gao et al., 2024; Liu et al., 2024) learn residuals to the pretrained weight matrices in the form of low-rank factorization, removing the added inference cost in adapter based methods. BitFit (Zaken et al., 2022) fine-tunes all the bias terms in a pretrained backbone. MEND (Mitchell et al., 2022) edits a pretrained model by learning fine-tuning weights from the gradient inputs with a low-rank MLP parameterization. FacT (Jie & Deng, 2023) shares the trainable parameters across Self-Attention and MLP blocks. While this reduces trainable parameters, it has a drawback: Transformer modules serve distinct functions - attention layers largely handle syntactical and in-context learning abilities (Bietti et al., 2023; Voita et al., 2019), while MLP layers encode factual knowledge (Meng et al., 2022). Sharing parameters across these modules may therefore be suboptimal. Tied-LoRA (Renduchintala et al., 2024) shares the residual weights across layers, and also across Query, Key and Value components. In Section 5.2, we show that the weight-awareness in WeGeFT is essential for parameter sharing across layers, enabling it to outperform Tied-LoRA.

**Hypernetworks.** (Ha et al., 2016) introduced Hypernet-

works, i.e., neural networks that generate the parameters for other neural networks, in language modeling tasks by generating the weights of an LSTM (Hochreiter & Schmidhuber, 1997). Hypernetworks have previously been applied for few-shot classification (Zhao et al., 2020; Zhmoginov et al., 2022), transfer learning (Requeima et al., 2019) and continual learning (von Oswald et al., 2020; Yin et al., 2022). Similar to our proposed approach, (Requeima et al., 2019) learns to adapt a global feature extractor through an adaptation network. In a few shot continual learning setup, (Vladymyrov et al., 2023) uses a hyper-Transformer to generate the parameters for a separate Convolutional Neural Network (ConvNet), which use as inputs both a support set of images of the current task and the ConvNet parameters generated for the previous tasks. HyperFormer++ (Mahabadi et al., 2021b) uses a Multi-Layer Perceptron (MLP) to generate the parameters from layer embedding and a latent vector for Adapters (Houlsby et al., 2019) introduced across layers of a pretrained model in a multitask setting. Unlike (Mahabadi et al., 2021b), we directly use the weights of the frozen pretrained model, thus eliminating the need for embeddings.

**Neural Functionals**: Our approach is related to neural functionals that aim to learn deep neural networks acting on the weights of other neural networks. For toy problems, equivariant architectures have been explored for tasks like classifying implicit neural representations (Navon et al., 2023; Zhou et al., 2023a;b; Kofinas et al., 2024), adapting model architectures to new domains (Navon et al., 2023), predicting model generalization performance (Zhou et al., 2023a;b; Kofinas et al., 2024; Lim et al., 2023), and learned optimizers (Zhou et al., 2024). However, our work is the first to explore fine-tuning of a model using it's own weights. We do not use equivariant architectures, but note that this direction of work is orthogonal to ours, and can be further explored in the future.

**Our Contributions** are summarized as follows:

- **Weight-Generative Fine-Tuning Framework:** WeGeFT introduces a novel formulation where fine-tuning weight residuals are parameterized directly using pretrained weights, leveraging their inherent knowledge for better expressivity, and leading to multiplicative updates in the

weight space for richer and more structured transformations compared to additive methods.

- **Multi-Faceted Efficiency:** WeGeFT achieves multi-faceted efficiency across parameters, representation, compute and memory by using shared low-rank matrices $(\phi, \psi)$, significantly reducing learnable parameters while retaining LoRA's simplicity without sacrificing performance (See Figure 2 and Figure 3).
- **Unified Parameter and Representation Efficiency:** WeGeFT bridges the gap between parameter-efficient and representation-efficient fine-tuning by unifying weight-generative parameterization and shared low-rank matrices.
- **Strong and Scalable Performance:** WeGeFT consistently matches or outperforms LoRA and its variants, achieving superior scalability with stronger pretrained models like Llama 3.

## 3. Approach

In this section, we elaborate on our simple formulation of WeGeFT (Eqn. 3) from the more general parameter-generation perspective, which can provide deeper insights.

### 3.1. Weight Generation for Explicit Weight-Awareness

For a layer $l \in L$ of a pretrained Transformer model, to learn its fine-tuning weights, denoted by $\Delta W^l$, that are aware of pretrained weights $W^l$ to "carry over" their knowledge, a general formulation is to train a generator network,

$$\Delta W^l = \mathbb{G}(W^l; \Theta), \forall l \in L, \qquad (5)$$

where $\Theta$ is the learnable parameters of the weight generator.

$\mathbb{G}(\cdot; \Theta)$ needs to be parameterized in a way to meet the desired multi-faceted efficiency. Inspired by the low-rank parameterization scheme in LoRA, we have,

$$\mathbb{G}(W^l; \Theta) = f_\psi \circ g_\theta \circ f_\phi(W^l), \forall l \in L, \qquad (6)$$

where both $f_\phi : \mathbb{R}^{d_{out} \times d_{in}} \rightarrow \mathbb{R}^{d_{out} \times r}$ and $f_\psi(x) : \mathbb{R}^{d_{out} \times r} \rightarrow \mathbb{R}^{d_{out} \times d_{in}}$ are linear projection layers (without bias terms) with $r$ representing the "rank". $g_\theta : \mathbb{R}^{d_{out} \times r} \rightarrow \mathbb{R}^{d_{out} \times r}$ realizes latent transformations in the low $r$-dim space. $\Theta = (\phi, \psi, \theta)$ collects all learnable parameters.

We note that this design offers a very flexibile way to capture underlying contingency between the fine-tuning weight residuals and the pretrained weights in all $L$ layers. Surprisingly, we observe that we do not need $g_\theta$ based on our ablation studies (see Section 5.1). In other words, $g_\theta$ is an identity transformation, leading to the simple formulation in Eqn. 3. Our (post-hoc) intuitive understanding is:

- If $g_\theta$ includes only linear transformations, it can be naturally absorbed into $f_\phi$ and/or $f_\psi$.
- If $g_\theta$ is an overall non-linear transformation, $\mathbb{G}(W^l; \Theta)$ applies the non-linear transformation in the weight space, which may not be necessary. After all, iterative updates

in weight space, including the from-scratch-training of the pretrained weights $W^l$ themselves, are mostly simple updates based on SGD. Nonlinear transformations may be destructive to the "carry over" knowledge in the pretrained weights, thus negatively impact weight-awareness. More importantly, with a nonlinear $g_\theta$, our WeGeFT will sacrifice the multiplicative updates in Eqn. 3, $\hat{W}^l = W^l \cdot (\mathbb{I} + \phi \cdot \psi)$, to additive updates, $\hat{W}^l = W^l + g_\theta(W^l \cdot \phi) \cdot \psi$, which will also significantly impact compute and memory efficiency.

**WeGeFT can be applied along the $d_{out}$ dimension too.** In Eqn. 3, $(\phi, \psi)$ are applied along the $d_{in}$ dimension of pretrained weights $W^l$. It is straightforward to apply WeGeFT along the $d_{out}$ dimension by,

$$\hat{W}^l = W^l + (\phi \cdot \psi)^\top \cdot W_l, \qquad (7)$$

where $\phi \in \mathbb{R}^{d_{out} \times r}$ and $\psi \in \mathbb{R}^{r \times d_{out}}$.

**WeGeFT Without Parameter Sharing.** It is straightforward to apply our WeGeFT (Eqn. 3 and Eqn. 7) *without* sharing $(\phi, \psi)$ across layers (denoted by WeGeFT-Sep), which will increase the learnable parameters of the counterpart (WeGeFT with parameter sharing), and to the same as LoRA. We show that WeGeFT-Sep can obtain on-par or better performance than LoRA, demonstrating the advantage of weight-awareness. This flexibility allows WeGeFT to scale more elegantly to larger and diverse datasets, which cannot be achieved by prior methods like VeRA and VB-LoRA.

### 3.2. WeGeFT as Token-Agnostic ReFT

Consider a linear layer with pretrained weights $W^l$ and the pretrained bias term $b^l$, and WeGeFT weights $\hat{W}^l = (\mathbb{I} + \phi \cdot \psi) \cdot W^l$ (Eqn. 3). For an input $x^l$, the output representation/activation at this layer is,

$$\hat{y}^l = x^l \cdot \hat{W}^{l\top} + b^l = \hat{x}^l \cdot W^{l\top} + b^l, \qquad (8)$$
$$\hat{x}^l = x^l \cdot (\mathbb{I} + \phi \cdot \psi)^\top,$$

where $\hat{x}^l$ is the "fine-tuned" input representation/activation using the same WeGeFT parameters. Hence, **our WeGeFT can be equivalently applied to the input activation, rather than the pretrained weights, to achieve the same fine-tuning effect**, maintaining the memory and compute efficiency of LoRA in implementation. Unlike the ReFT (Wu et al., 2024) that entails a dedicated search for where the representation interventions should apply at the token level, our WeGeFT eliminates the need of search, enabling token-agnosticity. Thanks to the parameter sharing, our WeGeFT can retain the representation efficiency.

## 4. Experiments

We conduct extensive experiments across Natural Language Generation and Visual Recognition, and compare our two-linear-layer parameterized WeGeFT with various PEFT

Table 1: Results of fine-tuning Llama 2 (7B) on the **Meta-MathQA** dataset and evaluating it on the **GSM8k** test set. All baseline results are obtained from (Wang et al., 2024), except VB-LoRA is trained by us together with our proposed WeGeFT. We train our WeGeFT and VB-LoRA by following the same settings in (Wang et al., 2024).

| Method | %Trainable | GSM8k (Acc.) |
|---|---|---|
| Full | 100 | $54.20_{\pm 0.42}$ |
| LoRA (Hu et al., 2022) | 0.297 | $42.08_{\pm 0.04}$ |
| PiSSA (Meng et al., 2024) | 0.297 | $44.54_{\pm 0.27}$ |
| rsLoRA (Kalajdzievski, 2023) | 0.297 | $45.62_{\pm 0.10}$ |
| LoRA+ (Hayou et al., 2024) | 0.297 | $52.11_{\pm 0.62}$ |
| DoRA (Liu et al., 2024) | 0.317 | $53.07_{\pm 0.75}$ |
| AdaLoRA (Zhang et al., 2023a) | 0.445 | $50.72_{\pm 1.39}$ |
| LoRA-GA (Wang et al., 2024) | 0.297 | $53.60_{\pm 0.30}$ |
| VB-LoRA (Li et al., 2024) | 1.194 | $44.93_{\pm 1.52}$ |
| Our WeGeFT$_{d_{in}}$ | **0.068** | $\mathbf{54.89}_{\pm 0.92}$ |

methods and ReFT. We also conduct ablation studies on the different parameterization schemes of WeGeFT. More details can be found in Appendix D. In all the experiments, we follow the baselines in selecting the layers to fine-tune for downstream tasks for fair comparisons. For clarity, we use WeGeFT$_{d_{in}}$ (Eqn. 3), WeGeFT$_{d_{out}}$ (Eqn. 7), and WeGeFT-Sep$_{d_{in}}$ (Eqn. 3 but without parameter sharing).

### 4.1. Arithmetic Reasoning

We demonstrate the multi-faceted efficiency of WeGeFT with experiments on the Math10k benchmark (Hu et al., 2023) for Arithmetic Reasoning, which is a small scale dataset enabling comprehensive evaluations. We conduct further experiments with MetaMathQA (Yu et al., 2024), a larger and higher quality fine-tuning dataset, to understand the impact of higher quality data on WeGeFT. In both experiments, we evaluate the model on the final answer following the same protocol used in prior works. Experimental details and hyperparameters are provided in Appendix D.2.

**Results**: Table 1 shows the results of finetuning using the MetaMathQA datasets, and evaluating on the GSM8k test set. WeGeFT outperforms all the prior methods with 4 times fewer parameters, and slightly outperform the full fine-tuning. We use the relatively smaller Math10k benchmark (Table 2) to run comprehensive evaluations for WeGeFT with respect to multi-faceted efficiency.

• **Parameter Efficiency**: WeGeFT efficiently adapts both strong and weak models, as shown by finetuning LLaMA-1 and LLaMA-2 (7B) on the Math10k benchmark in Table 2. LLaMA-1 (7B), a weak model for arithmetic reasoning (11% zero-shot accuracy on GSM8k (Touvron et al., 2023a)), requires substantial adaptation. Table 2 shows WeGeFT adapts LLaMA-1 (7B) far more effectively than prior PEFT methods. Reducing parameters in LoRA and DoRA significantly degrades performance—LoRA's accuracy drops from 50.9 ($r = 16$) to 48.9 ($r = 2$). At a com-

Table 2: Results of fine-tuning Llama 1 and 2 (7B) on the **Math10k** benchmark. The Mem. refers to GPU memory, and Wall Time is the time required to complete 1 epoch of training. All results are obtained by us using our code base for fair comparisons, except those by DiReFT and LoReFT using LLaMA 1 are from (Wu et al., 2024).

| | Method | Params (%) | Mem. (GB) | Wall Time | AQuA | GSM8k | MAWPS | SVAMP | Avg. Acc. |
|---|---|---|---|---|---|---|---|---|---|
| **LLaMA-1 (7B)** | LoRA$^{r=16}$ | 0.416 | 18.01 | 0.43 | 23.5 | 38.5 | 85.3 | 56.4 | **50.9** |
| | DoRA$^{r=16}$ | 0.427 | 20.37 | 2.36 | 21.5 | 37.9 | 86.0 | 55.3 | 50.2 |
| | WeGeFT-Sep$_{d_{in}}$ | 0.416 | 18.01 | 0.46 | 23.8 | 37.9 | 84.5 | 54.2 | 50.1 |
| | LoRA$^{r=2}$ | 0.052 | 17.74 | 0.43 | 23.1 | 34.6 | 83.9 | 54.1 | 48.9 |
| | DoRA$^{r=2}$ | 0.065 | 20.09 | 2.36 | 21.1 | 34.6 | 84.0 | 53.8 | 48.4 |
| | VeRA | 0.042 | 20.65 | 9.01 | 21.3 | 34.0 | 82.8 | 50.7 | 47.2 |
| | FacT-TT | 0.051 | 17.74 | 0.52 | 21.5 | 30.7 | 80.3 | 50.3 | 45.7 |
| | FacT-TK | 0.062 | 17.75 | 0.59 | 21.3 | 34.8 | 82.2 | 51.9 | 47.5 |
| | VB-LoRA | 0.840 | 18.33 | 0.42 | 21.26 | 29.3 | 78.9 | 49.5 | 44.7 |
| | WeGeFT$_{d_{in}}$ | 0.052 | 17.74 | 0.51 | 24.3 | 36.5 | 82.4 | 56.9 | **50.0** |
| | DiReFT | 0.031 | 31.42 | 0.26 | 21.3 | 24.1 | 74.5 | 42.7 | 40.6 |
| | LoReFT | 0.031 | 55.42 | 0.29 | 21.4 | 26.0 | 76.2 | 46.8 | 42.6 |
| | WeGeFT$_{d_{out}}$ | 0.016 | 17.71 | 0.36 | 20.74 | 33.0 | 80.8 | 53.5 | **47.0** |
| **Llama 2 (7B)** | LoRA$^{r=16}$ | 0.416 | 18.01 | 0.43 | 24.5 | 43.4 | 86.1 | 57.2 | 52.8 |
| | DoRA$^{r=16}$ | 0.429 | 20.37 | 2.35 | 24.1 | 41.4 | 87.1 | 57.1 | 52.4 |
| | WeGeFT-Sep$_{d_{in}}$ | 0.416 | 18.01 | 0.46 | 26.1 | 42.4 | 85.9 | 58.6 | **53.3** |
| | LoRA$^{r=2}$ | 0.052 | 17.74 | 0.42 | 24.7 | 40.2 | 85.0 | 56.0 | 51.5 |
| | DoRA$^{r=2}$ | 0.065 | 20.09 | 2.35 | 24.0 | 40.6 | 84.6 | 56.0 | 51.3 |
| | VeRA | 0.042 | 20.65 | 9.00 | 23.5 | 38.7 | 85.3 | 54.3 | 50.4 |
| | VB-LoRA | 0.840 | 18.33 | 0.43 | 22.4 | 33.4 | 81.4 | 52.4 | 47.4 |
| | FacT-TT | 0.051 | 17.74 | 0.52 | 24.9 | 38.3 | 81.9 | 56.2 | 50.3 |
| | FacT-TK | 0.062 | 17.75 | 0.59 | 24.5 | 41.0 | 85.7 | 54.4 | 51.4 |
| | WeGeFT$_{d_{in}}$ | 0.052 | 17.74 | 0.50 | 23.6 | 42.4 | 84.2 | 57.4 | **51.9** |
| | DiReFT | 0.031 | 31.42 | 0.26 | 20.5 | 27.9 | 77.5 | 45.9 | 42.9 |
| | LoReFT | 0.031 | 55.42 | 0.29 | 24.8 | 31.7 | 79.6 | 50.9 | 46.7 |
| | WeGeFT$_{d_{out}}$ | 0.016 | 17.71 | 0.39 | 26.1 | 38.0 | 83.1 | 57.3 | **51.1** |

parable parameter count ( 0.05%), WeGeFT outperforms all baselines (LoRA, DoRA, VeRA, VB-LoRA) and nearly matches LoRA at $r = 16$. **This demonstrates that generating fine-tuning residuals from pretrained weights improves parameter efficiency by enabling adaptation with minimal parameters, whereas LoRA and DoRA with $r = 2$ struggle to perform well**.

• **Computational and Memory Efficiency**: *WeGeFT maintains compute and memory efficiency while achieving equal or higher accuracy than prior PEFT methods, unlike DoRA and VeRA, which compromise efficiency.* Although VeRA reduces trainable parameters, it requires a large intermediate dimension for fixed random weights (12288 here). On Math10k, VeRA takes ∼9 hours/epoch and 20.65GB GPU memory, whereas LoRA and WeGeFT need only ∼0.5 hours/epoch and 17.74GB under the same setup. Table 10 in Appendix A confirms that reducing the intermediate dimension to 1024 (as in (Kopiczko et al., 2023)) lowers time and memory costs but causes a severe accuracy drop. Table 11 in Appendix B shows that WeGeFT is efficient even with mixed-precision float16. Thus, Similar to LoRA, WeGeFT can be used even with consumer GPUs.

• **Impact of Data Quality**: WeGeFT "reacts" positively to the quality of training datasets, as can be seen by the relative

Table 3: Results of fine-tuning LLaMA-1 7B, Llama 2 7B and Llama 3 8B on eight Commonsense Reasoning benchmarks (**Commonsense170k**). ReFT results are obtained from (Wu et al., 2024).

| | Method | Params (%) | BoolQ | PIQA | SIQA | HellaS. | WinoG. | ARC-e | ARC-c | OBQA | Avg |
|---|---|---|---|---|---|---|---|---|---|---|---|
| **LLaMA-1 (7B)** | LoRA (Hu et al., 2022) | 0.416 | 73.3 | 84.5 | 80.4 | 94.2 | 85.5 | 87.6 | 72.6 | 85.6 | 83.0 |
| | DoRA (Liu et al., 2024) | 0.427 | 73.4 | 84.8 | 80.7 | 94.4 | 85.7 | 87.4 | 72.4 | 85.9 | **83.1** |
| | VeRA (Kopiczko et al., 2023) | 0.023 | 70.4 | 82.4 | 79.9 | 91.4 | 81.8 | 83.3 | 67.0 | 80.6 | 79.6 |
| | VB-LoRA (Li et al., 2024) | 0.840 | 70.5 | 82.6 | 79.3 | 92.5 | 83.1 | 84.5 | 68.1 | 81.7 | 80.3 |
| | WeGeFT$_{d_{in}}$ | 0.052 | 72.8 | 84.7 | 80.8 | 93.9 | 84.3 | 86.7 | 72.3 | 85.1 | 82.6 |
| | DiReFT (Wu et al., 2024) | 0.031 | 69.5 | 83.0 | 79.0 | 92.5 | 80.5 | 82.2 | 68.0 | 77.5 | 79.0 |
| | LoReFT (Wu et al., 2024) | 0.031 | 69.3 | 84.4 | 80.3 | 93.1 | 84.2 | 83.2 | 68.2 | 78.9 | 80.2 |
| | WeGeFT$_{d_{out}}$ | 0.016 | 71.5 | 83.4 | 81.1 | 93.6 | 83.7 | 86.1 | 72.0 | 83.9 | **81.9** |
| **Llama 2 (7B)** | LoRA (Hu et al., 2022) | 0.416 | 74.8 | 85.9 | 80.8 | 94.8 | 86.3 | 88.3 | 75.4 | 85.9 | 84.0 |
| | DoRA (Liu et al., 2024) | 0.427 | 74.6 | 86.2 | 81.1 | 94.9 | 86.8 | 89.1 | 75.9 | 86.7 | **84.4** |
| | VeRA (Kopiczko et al., 2023) | 0.023 | 71.9 | 82.2 | 80.0 | 92.2 | 83.3 | 84.3 | 68.8 | 80.5 | 80.4 |
| | VB-LoRA (Li et al., 2024) | 0.840 | 71.6 | 83.2 | 79.7 | 92.8 | 83.5 | 84.8 | 69.3 | 81.9 | 80.8 |
| | WeGeFT$_{d_{in}}$ | 0.052 | 73.9 | 85.7 | 82.1 | 94.6 | 85.6 | 88.3 | 74.7 | 85.4 | 83.8 |
| | DiReFT (Wu et al., 2024) | 0.031 | 70.8 | 83.6 | 80.2 | 93.6 | 82.1 | 84.8 | 70.4 | 81.5 | 80.9 |
| | LoReFT (Wu et al., 2024) | 0.031 | 71.1 | 83.8 | 80.8 | 94.3 | 84.5 | 85.6 | 72.2 | 82.3 | 81.8 |
| | WeGeFT$_{d_{out}}$ | 0.016 | 73.4 | 85.2 | 81.8 | 94.3 | 85.3 | 87.7 | 74.9 | 83.8 | **83.3** |
| **Llama 3 (8B)** | LoRA (Hu et al., 2022) | 0.392 | 74.6 | 89.4 | 81.3 | 95.9 | 87.7 | 91.9 | 81.7 | 87.7 | 86.3 |
| | DoRA (Liu et al., 2024) | 0.361 | 76.2 | 90.8 | 82.1 | 96.7 | 89.0 | 93.5 | 83.4 | 89.1 | 87.6 |
| | VeRA (Kopiczko et al., 2023) | 0.018 | 71.6 | 85.7 | 80.7 | 93.8 | 85.2 | 87.6 | 75.6 | 84.1 | 83.0 |
| | VB-LoRA (Li et al., 2024) | 0.712 | 72.3 | 83.7 | 80.0 | 93.1 | 84.1 | 85.1 | 70.5 | 81.9 | 81.3 |
| | WeGeFT$_{d_{in}}$ | 0.049 | 76.0 | 89.7 | 83.1 | 96.7 | 89.1 | 93.0 | 84.4 | 89.8 | **87.7** |
| | DiReFT (Wu et al., 2024) | 0.026 | 73.4 | 88.7 | 81.0 | 95.6 | 85.5 | 91.8 | 81.8 | 85.4 | 85.4 |
| | LoReFT (Wu et al., 2024) | 0.026 | 75.1 | 90.2 | 82.0 | 96.3 | 87.4 | 92.4 | 81.6 | 87.5 | 86.6 |
| | WeGeFT$_{d_{out}}$ | 0.013 | 75.7 | 89.9 | 82.5 | 96.4 | 88.7 | 92.5 | 82.3 | 86.3 | **86.8** |

performance improvement from Math10k benchmark (Table 2) to the MetaMathQA benchmark (Table 1).

• **Comparison with ReFT**: To compare with DiReFT and LoReFT, we apply WeGeFT$_{d_{out}}$ following their fine-tuning strategy (Eqn. 2). WeGeFT achieves higher average accuracy than both ReFT variants while using half the parameters on LLaMA-1 and LLaMA-2. ReFT's higher memory usage (and lower wall time) is due to its implementation lacking gradient checkpointing, whereas our HuggingFace PEFT-based implementation uses checkpointing for better scalability with large models.

## 4.2. Commonsense Reasoning

We use combined training data of eight benchmarks (i.e., **Commonsense170k**, containing a total of 170k training samples), and evaluate on their test sets individually, following the same protocol used in (Hu et al., 2023) and (Wu et al., 2024). The examples in the Commonsense170k are formulated as multiple choice questions and consist of BoolQ (Clark et al., 2019), PIQA (Bisk et al., 2020), SIQA (Sap et al., 2019), HellaSwag (Zellers et al., 2019), WinoGrande (Sakaguchi et al., 2021), Arc-e and Arc-c (Clark et al., 2018), and OBQA (Mihaylov et al., 2018) datasets. We experiment with LLaMA-1 (7B), Llama 2 (7B) and Llama 3 (8B) models. Experimental details including hyperparameters are provided in Appendix D.3.

**Results** are shown in Table 3. Based on the observations

of our initial experiments, WeGeFT-Sep does not show significant improvement over WeGeFT. Hence, we focus on evaluating WeGeFT on the Commonsense170k benchmark. We summarize the observations as follows:

• **The weight-awareness of our WeGeFT is positively correlated with the expressivity of the pretrained models.** WeGeFT slightly outperforms DoRA in fine-tuning Llama 3 (8B) using 8x fewer parameters, while DoRA outperforms WeGeFT in fine-tuning both LLaMA 1 and Llama 2 at the expense of 8x more parameters and more expensive training cost. Considering the trend of increasingly powerful pretrained large foundation models, WeGeFT shows a very promising potential due to its efficiency and strong performance.

• Although VeRA uses less parameters, its performance is much worse and the training cost is very high, similar to the observations on Math10k. Furthermore, our WeGeFT$_{d_{out}}$ outperforms VeRA with even less parameters and much more efficient training.

• Compared with DiReFT and LoReFT, our WeGeFT is still better and reduces the parameters by half.

## 4.3. Instruction Following

We fine-tune LLaMA-2 (7B) on a 52k subset of the WizardLM dataset (Xu et al., 2024), filtering out samples containing "Sorry" and "As an AI" following (Wang et al., 2024). We evaluate the instruction following ability on the

Table 4: Results of fine-tuning Llama 2 (7B) on the **WizardLM** dataset (Xu et al., 2024) and evaluating it on the **MT-Bench** (Zheng et al., 2023). All the results except VB-LoRA are obtained from (Wang et al., 2024). We train VB-LoRA and WeGeFT following the same settings as (Wang et al., 2024).

| Method | Params (%) | First Turn Score |
|---|---|---|
| Full | 100 | $5.56_{\pm 0.09}$ |
| LoRA (Hu et al., 2022) | 0.297 | $5.61_{\pm 0.10}$ |
| PiSSA (Meng et al., 2024) | 0.297 | $5.30_{\pm 0.02}$ |
| rsLoRA (Kalajdzievski, 2023) | 0.297 | $5.25_{\pm 0.03}$ |
| LoRA+ (Hayou et al., 2024) | 0.297 | $5.71_{\pm 0.08}$ |
| DoRA (Liu et al., 2024) | 0.317 | $\mathbf{5.97}_{\pm 0.02}$ |
| AdaLoRA (Zhang et al., 2023a) | 0.445 | $5.57_{\pm 0.05}$ |
| LoRA-GA (Wang et al., 2024) | 0.297 | $5.95_{\pm 0.16}$ |
| VB-LoRA (Li et al., 2024) | 1.194 | $5.57_{\pm 0.05}$ |
| WeGeFT$_{d_{in}}$ | **0.068** | $5.75_{\pm 0.11}$ |

MT-Bench dataset (Zheng et al., 2023), which spans domains such as math, roleplay, reasoning, and coding. We report the the single turn score by prompting an LLM judge (**GPT4**) to rate the responses from the fine-tuned model from 1-10. Experimental settings and hyperparameters can be found in Appendix D.4).

**Results** are shown in Table 4. Our WeGeFT is on-par with LoRA-GA and DoRA with 4 times fewer parameters. It also slightly outperforms the full fine-tuning.

### 4.4. Code Generation

We fine-tune Llama 2 (7B) using the Code-Feedback dataset (Zheng et al., 2024), which is a multi-turn dataset containing execution and human feedback. We evaluate the fine-tuned models on HumanEval (Chen et al., 2021), containing Python problems evaluated for functional correctness. Experimental settings and hyperparameters are in Section D.5.

Table 5: Results of fine-tuning Llama 2 (7B) on the **Code-Feedback** dataset (Zheng et al., 2024) and evaluating it on the **HumanEval** (Chen et al., 2021). All results except VB-LoRA are obtained from (Wang et al., 2024). We train VB-LoRA and WeGeFT following the same settings as (Wang et al., 2024).

| Method | Params (%) | Pass@1 |
|---|---|---|
| Full | 100 | $\mathbf{19.87}_{\pm 0.57}$ |
| LoRA (Hu et al., 2022) | 0.297 | $14.76_{\pm 0.17}$ |
| PiSSA (Meng et al., 2024) | 0.297 | $16.02_{\pm 0.78}$ |
| rsLoRA (Kalajdzievski, 2023) | 0.297 | $16.01_{\pm 0.79}$ |
| LoRA+ (Hayou et al., 2024) | 0.297 | $18.17_{\pm 0.52}$ |
| DoRA (Liu et al., 2024) | 0.317 | $19.75_{\pm 0.41}$ |
| AdaLoRA (Zhang et al., 2023a) | 0.445 | $17.80_{\pm 0.44}$ |
| LoRA-GA (Wang et al., 2024) | 0.297 | $19.81_{\pm 1.46}$ |
| VB-LoRA (Li et al., 2024) | 1.194 | $14.92_{\pm 0.92}$ |
| WeGeFT$_{d_{in}}$ | **0.068** | $19.39_{\pm 0.68}$ |

**Results** are shown in Table 5. Our WeGeFT is on-par with LoRA-GA and DoRA.

### 4.5. Visual Recognition

**Data.** We evaluate WeGeFT on the VTAB-1k benchmark (Zhai et al., 2019) and the fine-grained visual classification (FGVC) benchmark containing Caltech-UCSD Birds (Wah et al., 2011), NABirds (Horn et al., 2015), Oxford Flowers (Nilsback & Zisserman, 2008), Stanford Cars (Gebru et al., 2017), and Stanford Dogs (Khosla et al., 2011).

**Models.** We use the ViT-B/16 architecture (Dosovitskiy et al., 2021) pretrained on ImageNet21k dataset (Deng et al., 2009) using a supervised objective, with the checkpoints from the timm package (Wightman, 2019). We apply LoRA and WeGeFT to the output projection layers in MHSA, which is inspired by observations in (Savadikar et al., 2023). All hyperparameters are provided in Appendix D.6.

**Results:** Tables 6 and 7 show that our WeGeFT performs better than other PEFT methods on both FGVC, while using fewer parameters. The GPU memory consumption is similar among the different methods with negligible differences. With 5.9 times less parameters used (0.025M vs 0.147M), on FGVC tasks, our WeGeFT improves LoRA by 0.68% Top-1 accuracy.

Table 6: Results on the finegrained visual classification (FGVC) tasks with ViT-B/16 pretrained on ImageNet21k. The number of trainable parameters are reported without the classification head (which has the same number of parameters for all the methods).

| Method | Params (M) | CUBS | Bird | Flower | Dog | Car | Avg |
|---|---|---|---|---|---|---|---|
| VPT | 0.046 | 87.88 | 84.79 | 98.98 | 84.51 | 82.89 | 87.81 |
| BitFit | 0.083 | 87.75 | 84.61 | **99.32** | 85.23 | 84.01 | 88.18 |
| LoRA | 0.147 | 88.00 | 84.94 | **99.32** | 85.36 | **85.92** | 88.71 |
| WeGeFT$_{d_{in}}$ | **0.025** | **89.71** | **86.28** | 99.22 | **87.44** | 84.28 | **89.39** |

Table 7: Results on the VTAB benchmark (Zhai et al., 2019) with ViT-B/16 pretrained on ImageNet21k. Trainable parameters are reported the same way as Table 6.

| Method | Params (M) | Natural | Specialized | Structured | Avg |
|---|---|---|---|---|---|
| VPT | 0.046 | 81.0 | 85.7 | 58.9 | 72.7 |
| BitFit | 0.083 | 81.8 | 85.2 | 57.8 | 72.4 |
| LoRA | 0.147 | **82.0** | 85.9 | 61.0 | 74.0 |
| FacT-TT | 0.040 | 79.8 | 86.0 | 58.0 | 71.9 |
| FacT-TK | 0.069 | 80.0 | **86.8** | 60.9 | 73.4 |
| WeGeFT$_{d_{in}}$ | 0.025 | **82.0** | 86.3 | **61.1** | **74.1** |

## 5. Ablation Studies

### 5.1. Different Parameterization Schema for WeGeFT

As mentioned in Section 3.1, a simple linear transformation of the pretrained weights works surprisingly well in generating fine-tuning residual weights. To verify effects of non-linear $g_\theta()$ in Eqn. 6. We compare,

- *Transformer*: We treat the set of shared pretrained weights

across $L$ layers as a batch of $L$ sequences each consisting of $d_{out}$ tokens in $r$-dim space (after the first linear project layer $f_\phi$), denoted by $\mathcal{W}$. We then apply a single Transformer block (Vaswani et al., 2017).

- *MLP-Mixers*: Similar to vanilla Transformers, we apply a single MLP-Mixer (Tolstikhin et al., 2021) block.
- *Multi-Layer Perceptrons (MLPs)*: e.g., $g(\mathcal{W}; \theta) = \text{Linear}(\text{GELU}(\text{Linear}(\mathcal{W}; \theta_1)); \theta_2)$, where $\theta_1 \in \mathbb{R}^{m \cdot r \times r + m \cdot r}$ and $\theta_2 \in \mathbb{R}^{r \times m \cdot r + r}$ consist of weights and bias terms of the two linear layers with the MLP latent dimension ratio $m$ (e.g., $m = 2$).
- *Element-wise non-linearity functions* without learnable parameters (i.e., $\theta = \emptyset$): e.g., $g(\mathcal{W}) = \text{Sigmoid}(\mathcal{W})$ or $g(\mathcal{W}) = \text{GELU}(\mathcal{W})$.

Table 8: Comparisons between various non-linear transformations for $g_\theta$ on the FGVC benchmark.

| Schema | Params (M) | CUBS | Bird | Flower | Dog | Car | *Avg* |
|---|---|---|---|---|---|---|---|
| Identity | 0.025 | **89.71** | **86.28** | 99.22 | **87.44** | 84.28 | **89.39** |
| Sigmoid | 0.025 | 89.56 | 84.61 | 99.20 | 86.69 | 84.04 | 88.82 |
| GeLU | 0.025 | 89.70 | 85.30 | 99.19 | 86.71 | 83.81 | 88.94 |
| MLP | 0.036 | 89.06 | 85.44 | **99.30** | 86.17 | 84.24 | 88.84 |
| Transformer | 0.027 | 89.56 | 86.23 | 99.24 | 86.31 | 84.26 | 89.12 |
| MLP Mixer | 0.125 | 88.76 | 86.21 | 99.25 | 86.35 | **85.66** | 89.25 |

Through ablation studies on the FGVC benchmark, we verify that using any non-linear transformation for $g_\theta$ results in degradation in performance. We use the same settings as Section 4.5. As seen from Table 8, the simple two-linear layer formulation achieves better or equivalent performance than all other schema at a lower parameter cost. While we do not have a theoretical understanding yet, we hypothesize that the superior performance of the identity operation over more complex and non-linear operations is because of difficulty in optimization. We only study the non-linear formulation on small models on simpler tasks due to computational constraints, and note that this presents an interesting avenue for future research.

### 5.2. Alternative Formulation of Tied LoRA

Tied LoRA (Renduchintala et al., 2024) uses a sophisticated design of sharing weights across layers. We test a straightforward parameter sharing LoRA, i.e., $\Delta W^l = B \cdot A \ \forall l \in L$, where $(B, A)$ is shared across layers. Table 9 shows that this strategy leads to much lower performance than our WeGeFT, which justifies the advantage of weight-awareness.

Table 9: Comparisons of Shared LoRA and WeGeFT on eight commonsense reasoning benchmarks.

| | Method | Params (%) | Avg |
|---|---|---|---|
| LLaMA-1 (7B) | Shared LoRA | 0.052 | 78.0 |
| | WeGeFT$_{d_{in}}$ | 0.052 | **82.6** |
| Llama-2 (7B) | Shared LoRA | 0.052 | 78.3 |
| | WeGeFT$_{d_{in}}$ | 0.052 | **83.8** |
| Llama-3 (8B) | Shared LoRA | 0.044 | 76.1 |
| | WeGeFT$_{d_{in}}$ | 0.044 | **87.7** |

## 6. Remarks on the Effectiveness of WeGeFT

Based on the experimental results, we may draw intuitive and potentially deeper understanding of PEFT and ReFT methods using pretrained Transformer backbones: Pretrained Transformer backbones "distill" general and diverse knowledge from a large-scale pretraining dataset, encoded in the pretrained weights. When fine-tuning them at a downstream task, to "absorb" new information in the training data of the downstream task, *one of the simplest updates that minimally "distorts" and maximally "preserves" the pretrained knowledge* is defined by Eqn. 3 or Eqn. 7, thanks to the low-rank factorized linear projection in the parameter space. The newly "absorbed" information from the downstream task is also linearly expressed in the space spanned by the pretrained weights (knowledge).

## 7. Conclusion

We present Weight-Generative Fine-Tuning (WeGeFT) for adapting pretrained Transformer backbones on downstream tasks. Our WeGeFT learns to generate the fine-tuning weight-residuals for layers selected in fine-tuning directly from their frozen pretrained weights. It is parameterized using two-linear-layers (without bias terms). It achieves multi-faceted efficiency across parameters, representations, compute and memory in comparisons with LoRA and its variants, and ReFT. We conduct experiments across various tasks, including Natural Language Generation (instruction following, commonsense reasoning, code generation, and arithmetic reasoning), and Visual Recognition. WeGeFT shows strong performance while retaining multi-faceted efficiency.

## Impact Statement

This paper presents work whose goal is to advance the field of Machine Learning. There are many potential societal consequences of our work, none of which we feel must be specifically highlighted here.

## Acknowledgments

This research is partly supported by NSF IIS-1909644, ARO Grant W911NF1810295, ARO Grant W911NF2210010, NSF CMMI-2024688 and NSF IUSE-2013451. The views and conclusions contained herein are those of the authors and should not be interpreted as necessarily representing the official policies or endorsements, either expressed or implied, of ARO, NSF or the U.S. Government. The U.S. Government is authorized to reproduce and distribute reprints for Governmental purposes not withstanding any copyright annotation thereon.

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

# Appendix

## A. Detailed analysis of Training Time

To show the advantage of WeGeFT over VeRA, we conduct further experiments by setting the intermediate rank in VeRA to be 1024 (as used in (Kopiczko et al., 2023)). Table 10 shows that while reducing the dimension lowers the training time and memory costs, it causes a severe drop in accuracy.

Table 10: Results of fine-tuning Llama 1 and 2 (7B) on the **Math10k** benchmark. The Mem. refers to GPU memory, and Wall Time is the time required to complete 1 epoch of training. All results are obtained by us using our code base for fair comparisons, except those by DiReFT and LoReFT using LlaMA 1 are from (Wu et al., 2024).

| | Method | Params (%) | Mem. (GB) | Wall Time | AQuA | GSM8k | MAWPS | SVAMP | Avg. Acc. |
|---|---|---|---|---|---|---|---|---|---|
| **LLaMA-1 (7B)** | LoRA$^{r=16}$ | 0.416 | 18.01 | 0.43 | 23.5 | 38.5 | 85.3 | 56.4 | **50.9** |
| | DoRA$^{r=16}$ | 0.427 | 20.37 | 2.36 | 21.5 | 37.9 | 86.0 | 55.3 | 50.2 |
| | WeGeFT-Sep$_{d_{in}}$ | 0.416 | 18.01 | 0.46 | 23.8 | 37.9 | 84.5 | 54.2 | 50.1 |
| | LoRA$^{r=2}$ | 0.052 | 17.74 | 0.43 | 23.1 | 34.6 | 83.9 | 54.1 | 48.9 |
| | DoRA$^{r=2}$ | 0.065 | 20.09 | 2.36 | 21.1 | 34.6 | 84.0 | 53.8 | 48.4 |
| | VeRA$^{r=12288}$ | 0.042 | 20.65 | 9.01 | 21.3 | 34.0 | 82.8 | 50.7 | 47.2 |
| | VeRA$^{r=1024}$ | 0.015 | 17.80 | 1.15 | 23.0 | 30.5 | 79.1 | 48.4 | 45.2 |
| | WeGeFT$_{d_{in}}$ | 0.052 | 17.74 | 0.51 | 24.3 | 36.5 | 82.4 | 56.9 | **50.0** |
| | WeGeFT$_{d_{out}}$ | 0.016 | 17.71 | 0.36 | 20.7 | 33.0 | 80.8 | 53.5 | **47.0** |
| **Llama 2 (7B)** | LoRA$^{r=16}$ | 0.416 | 18.01 | 0.43 | 24.5 | 43.4 | 86.1 | 57.2 | 52.8 |
| | DoRA$^{r=16}$ | 0.429 | 20.37 | 2.35 | 24.1 | 41.4 | 87.1 | 57.1 | 52.4 |
| | WeGeFT-Sep$_{d_{in}}$ | 0.416 | 18.01 | 0.46 | 26.1 | 42.4 | 85.9 | 58.6 | **53.3** |
| | LoRA$^{r=2}$ | 0.052 | 17.74 | 0.42 | 24.7 | 40.2 | 85.0 | 56.0 | 51.5 |
| | DoRA$^{r=2}$ | 0.065 | 20.09 | 2.35 | 24.0 | 40.6 | 84.6 | 56.0 | 51.3 |
| | VeRA$^{r=12288}$ | 0.042 | 20.65 | 9.00 | 23.5 | 38.7 | 85.3 | 54.3 | 50.4 |
| | VeRA$^{r=1024}$ | 0.015 | 17.80 | 1.15 | 23.6 | 35.5 | 82.1 | 53.3 | 48.6 |
| | WeGeFT$_{d_{in}}$ | 0.052 | 17.74 | 0.50 | 23.6 | 42.4 | 84.2 | 57.4 | **51.9** |
| | WeGeFT$_{d_{out}}$ | 0.016 | 17.71 | 0.39 | 26.1 | 38.0 | 83.1 | 57.3 | **51.1** |

## B. Performance of WeGeFT with mixed precision float16

We conduct additional experiments with LLaMA-1 (7B) using mixed-precision float16 instead of mixed-precision bfloat16. The table below shows that WeGeFT and LoRA experience a similar relative performance drop with float16 while maintaining comparable memory and wall-time, consistent with float16's known limitations as compared to bfloat16. The performance drop due to float16 can be offset by increasing the number of trainable parameters in WeGeFT. As shown in the table, WeGeFT with a rank of 128 outperforms LoRA even with float16, while using four times fewer parameters. These results further confirm WeGeFT's compatibility with any device that supports LoRA.

Table 11: Results of fine-tuning Llama 1 (7B) on the **Math10k** benchmark with pretrained weights and activations converted to float16 and bfloat16 precisions.

| | Method | Params (%) | Mem. (GB) | Wall Time | AQuA | GSM8k | MAWPS | SVAMP | Avg. Acc. |
|---|---|---|---|---|---|---|---|---|---|
| bfloat16 | LoRA$^{r=16}$ | 0.416 | 18.01 | 0.43 | 23.5 | 38.5 | 85.3 | 56.4 | **50.9** |
| | WeGeFT$_{d_{in}}^{r=64}$ | 0.052 | 17.74 | 0.51 | 24.3 | 36.5 | 82.4 | 56.9 | 50.0 |
| float16 | LoRA$^{r=16}$ | 0.416 | 18.07 | 0.42 | 21.8 | 37.9 | 84.7 | 57.1 | 50.4 |
| | WeGeFT$_{d_{in}}^{r=64}$ | 0.052 | 17.75 | 0.43 | 22.7 | 36.0 | 83.5 | 54.9 | 49.3 |
| | WeGeFT$_{d_{in}}^{r=128}$ | 0.104 | 17.80 | 0.43 | 22.7 | 38.5 | 84.6 | 56.1 | **50.5** |

## C. Analysis of WeGeFT vs. LoRA

To rigorously distinguish WeGeFT from LoRA, we perform an analytical gradient comparison in the context of fine-tuning Transformer-based architectures.

### C.1. Gradient Analysis

Consider a simplified Transformer layer with pretrained weights $W^l$ for layer $l$. LoRA fine-tunes by introducing additive low-rank residuals:

$$\hat{W}^l_{\text{LoRA}} = W^l + B^l A^l \tag{9}$$

where $B^l$ and $A^l$ are learnable low-rank matrices.

In contrast, consider Eqn. 3, WeGeFT fine-tunes through a multiplicative residual explicitly dependent on pretrained weights:

$$\hat{W}^l_{\text{WeGeFT}} = W^l(I + \phi \cdot \psi) \tag{10}$$

with shared low-rank matrices $\phi$ and $\psi$.

Let $\mathcal{L}$ denote a scalar loss function (e.g., cross-entropy). For LoRA, the gradient computations with respect to the matrices $A^l$ and $B^l$ are:

$$\frac{\partial \mathcal{L}}{\partial A^l} = (B^l)^\top X'^l, \quad \frac{\partial \mathcal{L}}{\partial B^l} = X'^l(A^l)^\top \tag{11}$$

where $X'^l = \left(\frac{\partial \mathcal{L}}{\partial X^l}\right)^\top X^{l-1}$ aggregates local gradient information.

For WeGeFT, gradients for $\phi$ and $\psi$ include information aggregated across layers due to parameter sharing:

$$\frac{\partial \mathcal{L}}{\partial \psi} = \phi^\top \sum_l (W^l)^\top X'^l, \quad \frac{\partial \mathcal{L}}{\partial \phi} = \sum_l (W^l)^\top X'^l \psi^\top \tag{12}$$

Here, WeGeFT gradients inherently integrate knowledge from pretrained weights across multiple layers, encapsulating broader contextual and structural dependencies than LoRA.

### C.2. Implications for Optimization Dynamics

- **Layer-wise vs. Global Updates:** LoRA updates parameters in isolation per layer, restricting interaction. In contrast, WeGeFT updates are global, considering inter-layer correlations and leading to more cohesive and stable optimization trajectories.
- **Pretrained Knowledge Utilization:** By explicitly multiplying residuals with pretrained weights, WeGeFT exploits existing model structure, preserving crucial pretrained information, potentially yielding superior convergence and generalization.
- **Expressivity and Efficiency Trade-off:** WeGeFT maintains expressivity through multiplicative updates despite substantial parameter sharing, balancing parameter efficiency without compromising learning capacity, unlike traditional additive methods such as LoRA.

### C.3. Summary of Advantages

- **Improved Parameter Efficiency:** Explicitly leverages pretrained weights to achieve stronger fine-tuning results with fewer learnable parameters.
- **Optimized Gradient Flow:** Gradients leverage global information, enabling coordinated fine-tuning across layers.

This analysis underpins the empirical advantages of WeGeFT observed in extensive experimentation, highlighting fundamental theoretical distinctions from LoRA.

# D. Implementation Details and Hyperparameter Tuning

In practice, we use a scaling factor of $\frac{\alpha}{r}$ for residuals as done in LoRA (Hu et al., 2022). We also use dropout (Srivastava et al., 2014) on the pretrained weights before transforming using WeGeFT parameters as a means of regularization:

$$\hat{W}^l = W^l + \frac{\alpha}{r}\text{Dropout}(W^l) \cdot \phi \cdot \psi, \tag{13}$$

We omit this in the main section for ease of notation and simplicity, as it does not affect the analysis. In experiments, we initialize $\psi$ to all zeros and $\phi$ to Kaiming Uniform initialization (He et al., 2015).

## D.1. Computing Resources and Code

All our experiments are run on a single Nvidia A100 GPU. Our code is provided in the supplementary materials.

## D.2. Arithmetic Reasoning

On the Math10k, we follow (Wu et al., 2024), and tune the hyperparameters by fine-tuning the LLaMA-1 (7B) model on the GSM8k dataset (Cobbe et al., 2021) using a separate validation set constructed from the training set, and use the same hyperparamters for Llama-2 (7B). Table 12 shows the hyperparameters used in our experiments. We perform hyperparameter search using the seed 42, and report the final results by averaging over three runs with seeds 42, 43, and 44. We use a greedy decoding scheme during inference, with a maximum new token number of 512. For experiments on fine-tuning Llama 2 (7B) on MetaMathQA and evaluating on GSM8k, we use the hyperparameters from (Wang et al., 2024), and tune the learning rate on a validation split from Meta-MathQA. We report average scores across 3 runs with seeds 42, 43, 44.

Table 12: Hyperparameters used for the Math10k experiments. We use greedy sampling following (Wu et al., 2024)

| | Hyperparameter | Value |
|---|---|---|
| | Max Sequence Length | 512 |
| | Optimizer | AdamW |
| | Weight Decay | 0.0 |
| | LR Scheduler | Linear |
| | Batch Size | 16 |
| | Epochs | 3 |
| WeGeFT$_{d_{in}}$ | Learning Rate | $4 \times 10^{-4}$ |
| | Rank | 64 |
| | Scaling Factor | 128 |
| | Warmup Ratio | 0.1 |
| | Dropout | 0.1 |
| | Fine-Tuned Layers | Query, Key, Value, Up Projection, Down Projection |
| WeGeFT$_{d_{out}}$ | Learning Rate | $7 \times 10^{-4}$ |
| | Rank | 64 |
| | Scaling Factor | 64 |
| | Warmup Ratio | 0.06 |
| | Fine-Tuned Layers | Out Projection, Down Projection |

## D.3. Commonsense Reasoning

We tune the hyperparameters for commonsense reasoning by fine-tuning the LLaMA-1 model on the BoolQ dataset (Clark et al., 2019) using a separate validation set constructed from the training set. Table 14 shows the hyperparameters used in our experiments. We search for the hyperparameters using LLaMa-1 (7B) and use the same hyperparameters for LLaMA-1 (13B), Llama 2 (7B) and Llama 3 (8B) models. We perform hyperparameter search using the seed 42, and report the final results by averaging over three runs with seeds 42, 43, and 44. We use a greedy decoding scheme during inference, with a maximum new token number of 32.

Table 13: Hyperparameters used for fine-tuning on MetaMathQA and evaluating on GSM8k.

| | Hyperparameter | Value |
|---|---|---|
| | Max Sequence Length | 1024 |
| | Optimizer | AdamW |
| | Weight Decay | 0.0 |
| | LR Scheduler | Cosine |
| | Batch Size | 32 |
| | Epochs | 1 |
| WeGeFT | Learning Rate | $5 \times 10^{-4}$ |
| | Rank | 64 |
| | Scaling Factor | 128 |
| | Warmup Ratio | 0.03 |
| | Dropout | 0.0 |
| | Fine-Tuned Layers | All linear layers (excluding vocabulary projection and head) |
| | Generation: Temperature | 0.8 |
| | Generation: top_p | 0.95 |

Table 14: Hyperparameters used for the commonsense reasoning experiments. We use greedy sampling following (Wu et al., 2024)

| | Hyperparameter | Value |
|---|---|---|
| | Max Sequence Length | 512 |
| | Optimizer | AdamW |
| | Weight Decay | 0.0 |
| | LR Scheduler | Linear |
| | Batch Size | 16 |
| | Epochs | 3 |
| WeGeFT | Learning Rate | $9 \times 10^{-5}$ |
| | Rank | 64 |
| | Scaling Factor | 128 |
| | Warmup Ratio | 0.1 |
| | Fine-Tuned Layers | Query, Key, Value, Up Projection, Down Projection |
| WeGeFT (Output) | Learning Rate | $6 \times 10^{-4}$ |
| | Rank | 64 |
| | Scaling Factor | 64 |
| | Warmup Ratio | 0.06 |
| | Dropout | 0.0 |
| | Fine-Tuned Layers | Out Projection, Down Projection |

### D.4. Instruction Following

For fine-tuning Llama 2 (7B) on WizardLM and evaluating on MT-Bench, we use the hyperparameters from (Wang et al., 2024) and use the same learning rate as MetaMathQA experiments. We report average scores across 3 runs with seeds 42, 43, 44.

### D.5. Code Generation

For fine-tuning Llama 2 (7B) on Code-Feedback dataset (Zheng et al., 2024) and evaluating on HumanEval, we use the hyperparameters from (Wang et al., 2024) and tune the learning rate on a separate validation split from Code-Feedback. We report average scores across 3 runs with seeds 42, 43, 44.

### D.6. FGVC Experiments

For all the experiments, we use ViT-B/16 model (Dosovitskiy et al., 2021), which contains 12 transformer blocks, each with 12 heads in the Multi-Head Self-Attention (MHSA) blocks, and a dimension of 768. We use checkpoints from the

Table 15: Hyperparameters used for fine-tuning on Code-Feedback and evaluating on HumanEval.

|  | Hyperparameter | Value |
|---|---|---|
|  | Max Sequence Length | 1024 |
|  | Optimizer | AdamW |
|  | Weight Decay | 0.0 |
|  | LR Scheduler | Cosine |
|  | Batch Size | 32 |
|  | Epochs | 1 |
| WeGeFT | Learning Rate | $4 \times 10^{-4}$ |
|  | Rank | 64 |
|  | Scaling Factor | 128 |
|  | Warmup Ratio | 0.03 |
|  | Dropout | 0.0 |
|  | Fine-Tuned Layers | All linear layers (excluding vocabulary projection and head) |
|  | Generation: Temperature | 0.8 |
|  | Generation: top_p | 0.95 |

Table 16: Hyperparameters used for fine-tuning on WizardLM and evaluating on MT-Bench.

| Hyperparameter | Value |
|---|---|
| Max Sequence Length | 1024 |
| Optimizer | AdamW |
| Weight Decay | 0.0 |
| LR Scheduler | Cosine |
| Batch Size | 32 |
| Epochs | 1 |
| Learning Rate | $5 \times 10^{-4}$ |
| Rank | 64 |
| Scaling Factor | 128 |
| Warmup Ratio | 0.03 |
| Dropout | 0.0 |
| Fine-Tuned Layers | All linear layers (excluding vocabulary projection and head) |
| Generation: Temperature | 0.8 |
| Generation: top_p | 0.95 |

model pretrained on the ImageNet21k (Deng et al., 2009) under the supervised training protocol provided by the timm package. For both VTAB and FGVC experiments, we use a hyperparameter search using the validation sets and use the training+validation data during the final run and report the results on the test sets. The hyperparameter search space used in our experiments in provided in Table 17. We use the same train, validation and test splits as (Shi et al., 2023), *except for* Stanford Cars dataset (Gebru et al., 2017). Due to the unavailability of the dataset from the original source, and the difference in the format of the data provided by the updated source, we create our own training and validation split (with the same number of images as (Shi et al., 2023)) and use the official testing split. We initialize $\phi$ with zeros and $\psi$ with Kaiming uniform initialization.

Table 17: Hyperparameter search space used for FGVC experiments. During the search, we use 25 epochs due to computational constraints, and use 100 epochs in the final run with the selected hyperparameters

|  | **Hyperparameter** | **Values** |
|---|---|---|
| BitFit | Learning Rate
Weight Decay | $1e^{-3}, 1.5e^{-3}, 2e^{-3}, 2.5e^{-3}, 5e^{-3}, 1e^{-2}$
0.0 |
| VPT | Learning Rate
Weight Decay
Num. Prompts | $1e^{-3}, 1.5e^{-3}, 2e^{-3}, 2.5e^{-3}, 5e^{-3}, 1e^{-2}$
0.0
5 |
| LoRA | Learning Rate
Weight Decay
Rank $r$ | $1e^{-3}, 1.5e^{-3}, 2e^{-3}, 2.5e^{-3}, 5e^{-3}, 1e^{-2}$
0.01, 0.001, 0.0001, 0.0
8 |
| WeGeFT | Learning Rate
Weight Decay
Rank $r$ | $1e^{-4}, 2.5e^{-4}, 5e^{-4}, 1e^{-3}, 2.5e^{-3}, 5e^{-3}$
0.01, 0.001, 0.0001, 0.0
16 |
|  | Optimizer
LR Scheduler
Warmup Epochs
Epochs
Batch Size | AdamW
Cosine
5
100
32 |

# E. Visual Inspection of Our Two-Linear-Layer Parameterized WeGeFT

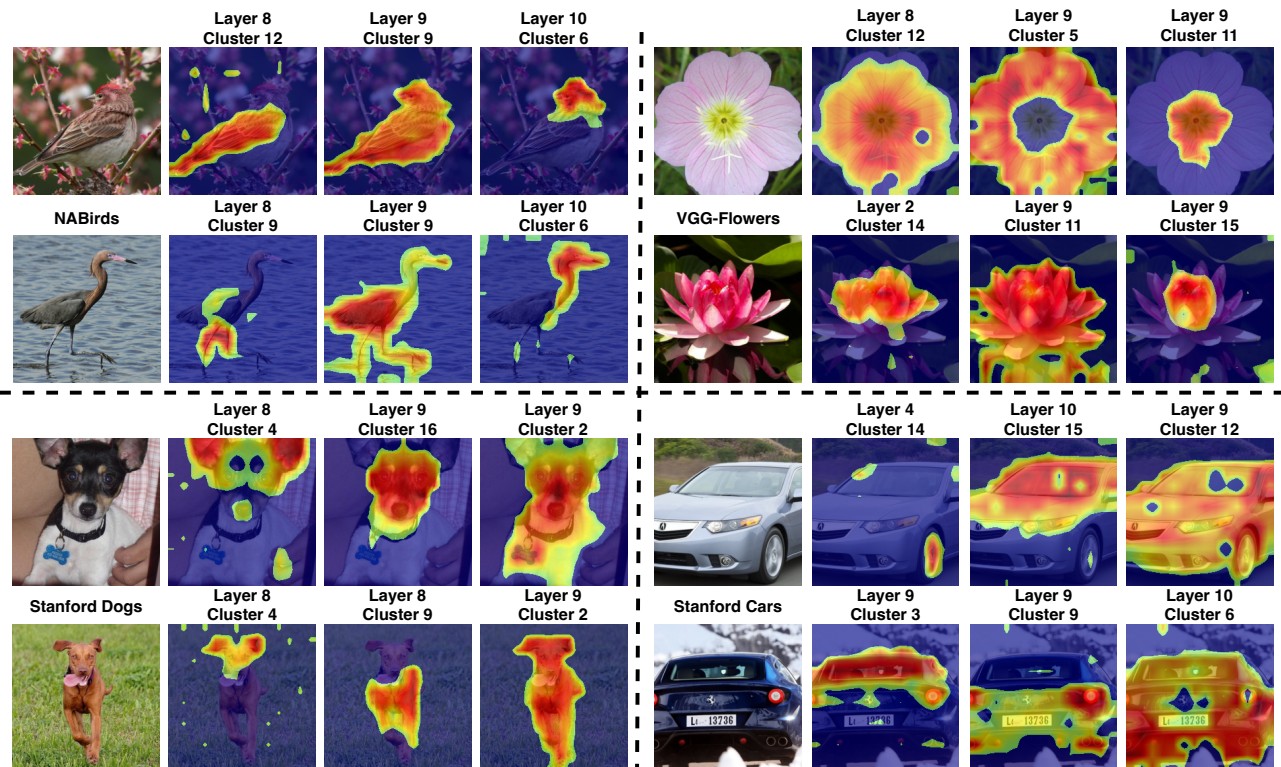

Figure 5: More examples of the visual interpretability of our two-linear-layer parameterized WeGeFT tested on the FGVC benchmark. We show examples of head, wings and legs of birds in the *top-left*, examples of flower petals in the *top-right*, examples of head, ears and legs of dogs in the *bottom-left*, and examples of tires, windshield and bumper of cars in the *bottom-right*.

