# OpenReview forum: "WeGeFT: Weight‑Generative Fine‑Tuning for Multi‑Faceted Efficient Adaptation of Large Models"
_ICML.cc/2025/Conference — ICML 2025 poster_

### Official Review · Reviewer_Zn72 · 2025-03-07

**Overall Recommendation:** 3

**Summary:**

The paper proposes Weight-Aware Fine-Tuning (WAFT) to generate fine-tuning weights directly from the pretrained weights.
Results show promising results on three datasets based on the LLaMA series of models.

**Claims And Evidence:**

Yes

**Essential References Not Discussed:**

No.

**Experimental Designs Or Analyses:**

Yes.
Table 1 shows the results of fine-tuning Llama 2 (7B) on the MetaMathQA dataset and evaluating it on the GSM8k test set, while Table 2 presents the results of fine-tuning Llama 1 and 2 (7B) on the Math10k benchmark.
In addition, Table 3 gives the results of fine-tuning LLaMA-1 7B, Llama 2 7B and Llama 3 8B on Commonsense170k.
The rationality of foundation model selection and benchmarking method is not clear.

**Methods And Evaluation Criteria:**

Yes

**Other Comments Or Suggestions:**

N/A

**Other Strengths And Weaknesses:**

Strengths:
1. Weight-Aware Fine-Tuning Framework is novel.
2. Experimental results are presented clearly.

Weaknesses:
1. Experimental designs, especially for using what foundation models and on what datasets, are not logically clear.
2. The claimed scalable performance is not validated by experiments.

**Questions For Authors:**

N/A

**Relation To Broader Scientific Literature:**

PEFT methods are meaningful for various applications based on foundation models.

**Theoretical Claims:**

No theoretical claims.

---

> ### Author Rebuttal · Authors · 2025-04-01
>
> Dear reviewer Zn72,
>
> Thank you for your feedback and efforts in reviewing our submission. We address your concerns as follow, which will be carefully incorporated in the revision.
>
> > **Experimental designs, especially for using what foundation models and on what datasets, are not logically clear**
>
> We follow prior works [1, 2, 3] in selecting foundation models and datasets, all of which are standard in the PEFT literature. We appreciate any suggestions on how to further clarify our choices.
>
> > **The claimed scalable performance is not validated by experiments**
>
> We have evaluated WAFT at scales of 7–8B parameters on large fine-tuning datasets for arithmetic reasoning, code completion, commonsense reasoning, and instruction following, all of which are tasks of practical interest. Based on these results, we believe our experiments demonstrate scalability. However, we welcome further insights on how to strengthen this validation.
>
> **References**
>
> [1] Zhiqiang Hu, Lei Wang, Yihuai Lan, Wanyu Xu, Ee-Peng Lim, Lidong Bing, Xing Xu, Soujanya Poria, Roy Ka-Wei Lee: LLM-Adapters: An Adapter Family for Parameter-Efficient Fine-Tuning of Large Language Models. EMNLP 2023: 5254-5276
>
> [2] Zhengxuan Wu, Aryaman Arora, Zheng Wang, Atticus Geiger, Dan Jurafsky, Christopher D. Manning, Christopher Potts: ReFT: Representation Finetuning for Language Models. NeurIPS 2024
>
> [3] Shaowen Wang, Linxi Yu, Jian Li: LoRA-GA: Low-Rank Adaptation with Gradient Approximation. NeurIPS 2024

---

### Official Review · Reviewer_M1uG · 2025-03-13

**Overall Recommendation:** 4

**Summary:**

The authors:
1. Identify current limitation of existing PEFT methods -- the replicated structures compromise time and memory efficiency.
2. Propose a novel approach that shares PEFT paratemers across layers.
3. Evaluate the proposed method on different testbed, including image classification, reasoning and generation tasks.
4. Perform ablation study on the effect of different latent operators g, ultimately finding that the identity operator is sufficient.

**Claims And Evidence:**

All of the authors’ claims are generally convincing.

**Essential References Not Discussed:**

See section "Questions For Authors".

**Experimental Designs Or Analyses:**

The experiments cover a wide range of tasks for foundation models, offering a comprehensive and sufficiently thorough evaluation.

**Methods And Evaluation Criteria:**

The evaluation benchmark is comprehensive. Providing convincing evidence for the performance of proposed method.

**Other Comments Or Suggestions:**

N/A

**Other Strengths And Weaknesses:**

Strenths:

A nice work overall.
1. The paper is well-written and well-organized.
2. The motivation is clear and proposed method is effective, easy to understand.
3. Experiment design are comprehensive and convincing.

Weaknesses:

See section "Questions For Authors".

**Questions For Authors:**

1. The concept of sharing information across PEFT layers appears very similar to approach proposed in [1]. I'm kind of superised the authors cite that work without comparison with them. I would be very interested in seeing a detailed comparison between these two mehods, particularly, how to explain if any difference in performance.
2. I'm interested in the performance of identify transformation conducted in section 5.1. Seems the authors present them with explanation. Do the authors have any hint on its superior performance versus other operators?

I don't mind raising my score if my questions (especially 1) are well-addressed.

[1] Jie, S. and Deng, Z. Fact: Factor-tuning for lightweight adaptation on vision transformer. In Williams, B., Chen, Y., and Neville, J.

**Relation To Broader Scientific Literature:**

The idea of describing a shared weights between PEFT layers is enlighting. I believe this methodology offers valuable insights for future PEFT research, specifically, enlights people on design of more effective and efficient architectures.

**Theoretical Claims:**

N/A

---

> ### Author Rebuttal · Authors · 2025-04-01
>
> Dear reviewer M1uG,
>
> Thank you for your feedback and efforts in reviewing our submission. We address your concerns as follow, which will be carefully incorporated in the revision.
>
> > **Comparison with FacT**
>
> We acknowledge that Factor Tuning (FacT) shares a similar parameter-sharing concept with WAFT. However, unlike WAFT, FacT shares parameters across both layers and modules (Self-Attention and MLP blocks). While this reduces trainable parameters, it has a drawback: Transformer modules serve distinct functions - attention layers largely handle syntactical and in-context learning abilities [1, 2], while MLP layers encode factual knowledge [3]. Sharing parameters across these modules may therefore be suboptimal.
>
> For a comprehensive evaluation, we compare FacT on the VTAB and Math10k benchmarks. We use the open-source implementation from the original paper for VTAB, employing the same model as in our FGVC experiments. For Math10k, we adapt FacT to the HuggingFace PEFT package and evaluate it on LLaMA-1 and LLaMA-2 (7B). The table below shows that WAFT outperforms both FacT variants on VTAB while using fewer parameters.
>
> Method|Params (M)|Natural|Specialized|Structured|Avg
> -|-|-|-|-|-
> VPT|0.046|81.0|85.7|58.9|72.7
> BitFit|0.083|81.8|85.2|57.8|72.4
> LoRA|0.147|**82.0**|85.9|61.0|74.0
> FacT-TT|0.040|79.8|86.0|58.0|71.9
> FacT-TK|0.069|80.0|**86.8**|60.9|73.4
> WAFT |0.025|**82.0**|86.3|**61.1**|**74.1**
>
> WAFT outperforms FacT on the Math10k benchmark, with a larger performance gap on LLaMA-1 than Llama 2. Since LLaMA-1 is weaker and requires more adaptation for arithmetic reasoning, this highlights WAFT’s ability to adapt both weak and strong models effectively, whereas FacT struggles with weaker models.
>
> Method|Params (%)|Mem. (GB)|Wall Time|AQuA|GSM8k|MAWPS|SVAMP|Avg
> -|-|-|-|-|-|-|-|-
> -|-|-|-|LLaMA 1|-|-|-|-
> FacT (TT) |0.051|17.74|0.52|21.5|30.7|80.3|50.3|45.7
> FacT (TK) |0.062|17.75|0.59|21.3|34.8|82.2|51.9|47.5
> WAFT|0.052|17.74|0.51|**24.3**|**36.5**|**82.4**|**56.9**|**50.0**
> -|-|-|-|Llama 2|-|-|-|-
> FacT (TT) |0.051|17.74|0.52|**24.9**|38.3|81.9|56.2|50.3
> FacT (TK) |0.062|17.75|0.59|24.5|41.0|**85.7**|54.4|51.4
> WAFT|0.052|17.74|0.50|23.6|**42.4**|84.2|**57.4**|**51.9**
>
> > **Effectiveness of the identity transformation**
>
> While we do not have a theoretical understanding yet, we hypothesize that the superior performance of the identity operation over more complex and non-linear operations is because of difficulty in optimization.
>
> ---
>
> **References**
>
> [1] Alberto Bietti, Vivien Cabannes, Diane Bouchacourt, Hervé Jégou, Léon Bottou: Birth of a Transformer: A Memory Viewpoint. NeurIPS 2023
>
> [2] Elena Voita, David Talbot, Fedor Moiseev, Rico Sennrich, Ivan Titov: Analyzing Multi-Head Self-Attention: Specialized Heads Do the Heavy Lifting, the Rest Can Be Pruned
>
> [3] Kevin Meng, David Bau, Alex Andonian, Yonatan Belinkov: Locating and Editing Factual Associations in GPT. NeurIPS 2022

---

> > ### Comment · Reviewer_M1uG · 2025-04-06
> >
> > Thanks the authors for addressing my concerns. i'll change my evaluation to accept.

---

### Official Review · Reviewer_NsTU · 2025-03-14

**Overall Recommendation:** 3

**Summary:**

1. Fine - tuning large pretrained Transformer models has two main focuses: parameter - efficient and representation - efficient fine - tuning. LoRA is a pioneering method, but its variants often sacrifice compute and memory efficiency or performance. ReFT is another approach, yet it has performance limitations.

2. The paper proposes Weight - Aware Fine - Tuning (WAFT), which generates fine - tuning weights from pretrained weights. It uses a simple low - rank formulation with two shared linear layers, aiming for multi - faceted efficiency in parameters, representations, compute, and memory.

3. WAFT is related to hypernetworks and neural functionals. It innovatively tunes models using their own weights. Its contributions include a novel framework, multi - faceted efficiency, unifying parameter and representation efficiency, and strong and scalable performance.
Experiments on arithmetic reasoning, commonsense reasoning, instruction following, code generation, and visual recognition show that WAFT outperforms many baseline methods. It can achieve better performance with fewer parameters while maintaining LoRA's compute and memory efficiency.

**Claims And Evidence:**

Yes.

**Essential References Not Discussed:**

No.

**Experimental Designs Or Analyses:**

Yes.

**Methods And Evaluation Criteria:**

Yes.

**Other Comments Or Suggestions:**

N.A.

**Other Strengths And Weaknesses:**

Novelty:

1. WAFT innovatively generates fine - tuning weights from pretrained weights. This is a new approach in fine - tuning Transformer models. It sets itself apart from LoRA and its variants by using weight - aware parameterization, which enables more effective utilization of pretrained knowledge. It unifies parameter - efficient and representation - efficient fine - tuning. By combining these two aspects, WAFT offers a more comprehensive solution. This unified approach simplifies the fine - tuning process and can potentially be applied to a wider range of tasks.
The design of WAFT, with two shared linear layers, is simple yet effective. It reduces the number of learnable parameters while maintaining or improving performance. This simplicity also contributes to its scalability and ease of implementation.

2. Experiments

The paper conducts extensive experiments across multiple tasks. It tests WAFT on arithmetic reasoning, commonsense reasoning, instruction following, code generation, and visual recognition. This wide range of tasks validates the method's generality and effectiveness in different scenarios. The experiments include comparisons with various baseline methods. By comparing WAFT with LoRA, DoRA, VeRA, and ReFT, among others, the paper clearly demonstrates WAFT's advantages. The results show that WAFT can achieve better performance with fewer parameters and maintain compute and memory efficiency.
Ablation studies are carried out to verify the effectiveness of WAFT's components. These studies test different parameterization schemes and alternative formulations. They help to understand the impact of each part of WAFT, providing insights into its design and performance.

3. Reference Integrity

The paper comprehensively reviews the related literature. It covers parameter - efficient fine - tuning methods, hypernetworks, and neural functionals. This shows a deep understanding of the field and places WAFT in the context of existing research.
All the references are properly cited. The authors use a wide range of sources, from conference papers to arXiv preprints. This gives credit to the original work and allows readers to further explore related research.
The reference list is up - to - date. It includes recent papers published in 2024, ensuring that the research builds on the latest developments in the field. This indicates the paper's relevance and timeliness.

Weakness

1. Limited Generalization Exploration
Although WAFT shows good performance on the tested tasks, its generalization to other architectures and tasks is not fully explored. The paper mainly focuses on Transformer - based models. It's unclear how well WAFT would work in models with different structures or in emerging applications.
There is a lack of analysis on the long - term stability of WAFT. As the field of deep learning evolves, new data and tasks may emerge. It's not clear if WAFT can maintain its performance and efficiency over time without further adjustments.
WAFT's performance in resource - constrained environments is not well - studied. The experiments are mostly conducted on a single Nvidia A100 GPU. It's unknown how the method would perform in devices with limited computational power and memory, which are common in real - world applications.

**Questions For Authors:**

N.A.

**Relation To Broader Scientific Literature:**

1. The paper's WAFT approach builds on the prior work of LoRA and its variants. While LoRA achieved efficiency in parameters, compute, and memory, its successors often traded off these aspects. WAFT addresses this by directly generating fine - tuning weights from pretrained weights, offering multi - faceted efficiency and better performance.

2. ReFT, another related method, focused on lightweight representation - editing modules. However, it had performance issues. WAFT unifies parameter - efficient and representation - efficient fine - tuning, improving upon ReFT by leveraging pretrained weight knowledge and achieving on - par or better performance.

**Theoretical Claims:**

Yes. Compete and correct proofs.

---

> ### Author Rebuttal · Authors · 2025-04-01
>
> Dear reviewer NsTU,
>
> Thank you for your feedback and efforts in reviewing our submission. We address your concerns as follows, which will be carefully incorporated in the revision.
>
> > **Performance in constrained environments**
>
> We use an A100 GPU to accelerate experiments, but the implementation is not tied to any specific GPU architecture. WAFT’s memory and wall-time requirements are comparable to or lower than LoRA, while VeRA and DoRA significantly increase both. Table 2 in the manuscript provides a detailed analysis, confirming WAFT’s suitability for any device that supports LoRA.
>
> For further verification, we conduct additional experiments with LLaMA-1 (7B) using mixed-precision float16 instead of mixed-precision bfloat16. The table below shows that WAFT and LoRA experience a similar relative performance drop with float16 while maintaining comparable memory and wall-time, consistent with float16’s known limitations as compared to bfloat16.
>
> The performance drop due to float16 can be offset by increasing the number of trainable parameters in WAFT. As shown in the table, WAFT with a rank of 128 outperforms LoRA even with float16, while using four times fewer parameters. **These results further confirm WAFT’s compatibility with any device that supports LoRA**.
>
>
> Method|Params (%)|Mem. (GB)|Wall Time|AQuA|GSM8k|MAWPS|SVAMP|Avg. Acc.
> -|-|-|-|-|-|-|-|-
> LoRA $^{r=16}$ (bfloat16)|0.416|18.01|0.43|23.5|38.5|85.3|56.4|50.9
> WAFT $^{r=64}$ (bfloat16)|0.052|17.74|0.51|24.3|36.5|82.4|56.9|50.0
> LoRA $^{r=16}$ (float16)|0.416|18.07|0.42|21.8|37.9|84.7|57.1|50.4
> WAFT $^{r=64}$ (float16)|0.052|17.75|0.43|22.7|36.0|83.5|54.9|49.3
> WAFT $^{r=128}$ (float16)|0.104|17.80|0.43|22.7|38.5|84.6|56.1|50.5
>
> > **Generalization to other architectures**
>
> We recognize the importance of exploring generalization. However, Transformers have proven to be highly scalable and flexible, and have emerged as the standard architecture due to their scalability and adaptability. Consequently, both our study and most prior PEFT research focus on Transformers. We believe extending WAFT to other architectures is a valuable future direction, and its strong performance in the context of prior research on fine-tuning Transformer models makes it a meaningful contribution.

---

### Official Review · Reviewer_UHSy · 2025-03-17

**Overall Recommendation:** 3

**Summary:**

This paper introduces Weight-Aware Fine-Tuning (WAFT), a novel parameter-efficient fine-tuning method for large pre-trained Transformer models. WAFT proposes to generate fine-tuning weights directly from the pre-trained weights using a low-rank formulation with shared linear layers across multiple transformer layers. By sharing parameters across layers, the authors claim that WAFT achieves multi-faceted efficiency in parameters, representations, compute, and memory, while maintaining or exceeding the performance of LoRA and its variants.
Experimental results are presented for various NLP tasks (commonsense reasoning, arithmetic reasoning, instruction following, code generation) as well as visual recognition to demonstrate the effectiveness of WAFT.

## update after rebuttal

I thank the authors for their rebuttal and willingness to improve clarity on the weight-aware term. I will maintain my score.

My 2) concern is indeed about having a $\mathbb{W}^l$ in the weight update that is not the pretrained weight which I suppose bears similarities to VeRA as suggested. There is not clear reason why this approach would not converge since VeRA does.

This study would help enhance the insights of WAFT to understand whether  the algorithm actually uses knowledge from the pretrained weight or if it simply uses properties of the learned matrix that arise during the pre-training stage (independence of features for example).

I suggest the authors include such study in the paper (or supplementary) if they obtain interesting results for the camera ready.

**Claims And Evidence:**

This paper proposes to modify LoRA's weight additive paradigm to a matrix-multiplication one. This is a valid idea to pursue although I do not feel that the weight-aware narrative pushed by the authors is very convincing.

I have two main concerns about the author's weight-aware claims:

(1) LoRA is weight-aware to an extent as the update is formulated as (W + BA) so the BA updates takes W in consideration when learning a solution. This is a bit hand-wavy from my part. Maybe the weight-aware term is not characteristic enough in this scenario which weakens the narative pushed by the paper to justify the multiplicative update.

(2) It is not always clear that the pre-trained weight are a good starting basis for finetuning. What about tasks where the zero-shot model is highly un-adapted ? In this case, starting without the pre-trained weights constraints may be an advantage. The cars dataset in Table 6 could be an example or any of the VTAB1k [1] specialized or structured subsets.

More exploration could be done on the relevance of using the pre-trained weight for the matrix multiplication, for example is a properly scaled random matrix performing close to the pretrained weights ?

[1] Zhai, Xiaohua, et al. "A large-scale study of representation learning with the visual task adaptation benchmark." ICLR 2020

**Essential References Not Discussed:**

SOTA is well represented

**Experimental Designs Or Analyses:**

Good design but more ablation study should have been conducted on using the pre-trained weight matrix for the multiplicative update.

**Methods And Evaluation Criteria:**

Good range of test datasets which adequately justify the performance of the proposed algorithm

**Other Comments Or Suggestions:**

In general the idea of multiplicative update is valid and appear to be effective experimentally.

The current justification as to why this is a good idea theoretically is not good enough as I do not find the weight-aware narrative compelling and ablations are missing on using other types of matrices for the multiplicative updates.

**Other Strengths And Weaknesses:**

Discussed above

**Questions For Authors:**

I was surprised of the 9 hours per epoch of VeRA line 307 which is almost 20 times longer than LoRA. Is this because of the very large intermediate dimension of 12k ? What about using a representation of 1024 as suggested in the original VeRA paper ?

**Relation To Broader Scientific Literature:**

PEFT is a relevant field, it is good to see contributions that veer away from LoRA's formulation.

**Theoretical Claims:**

Did not check the correctness of proofs

---

> ### Author Rebuttal · Authors · 2025-04-01
>
> Dear reviewer UHSy,
>
> Thank you for your feedback and efforts in reviewing our submission. We address your concerns (C1-C5) as follows, which will be carefully incorporated in the revision.
>
> > **C1: Weight-awareness in LoRA and WAFT**
>
> In general, any PEFT methods have to be (pretrained) weight-aware to be effective for downstream tasks. However, prior art like LoRA has `implicit` weight-awareness between the fine-tuning residual ($\Delta W^l = B^l\cdot A^l$) and the pretrained weights ($W^l$) via gradient backpropagation in learning (see Eqn.10 for a toy example in Appendix A). Our proposed WAFT harnesses `explicit` weight-awareness (in the forward computation) via learning $\Delta W^l = W^l\cdot \phi \cdot \psi$. In the abstract (lines 025-027), we define **WAFT, a novel approach that learns to generate fine-tuning weights directly from the pretrained wegiths**. (the generic formulation is in Sec.3.1). To address your concerns, we propose to use **generative weight-aware fine-tuning (GenWAFT)** to better highlight the characteristics of our proposed method in revision. The comprehensive experiments have verified its effectiveness.
>
> > **C2: Training time for VeRA**
>
> VeRA's long training time results from the large intermediate dimension used in our experiments. Since VeRA relies on fixed random matrices, a large dimension is necessary for reasonable accuracy. As shown in the table below, **reducing the dimension to 1024 (as in the original paper) significantly degrades performance**, despite lowering training time and parameter count. In contrast, our proposed WAFT maintains parameter and training efficiency while achieving comparable or superior performance to state-of-the-art methods.
>
> Method|Params (%)|Mem. (GB)|Wall Time|AQuA|GSM8k|MAWPS|SVAMP|Avg
> -|-|-|-|-|-|-|-|-
> -|-|-|-|LLaMA 1|-|-|-|-
> VeRA $^{r=12288}$ |0.042|20.65|9.01|21.3|34.0|82.8|50.7|47.2
> VeRA $^{r=1024}$ |0.015|17.80|1.15|23.0|30.5|79.1|48.4|45.2
> -|-|-|-|Llama 2|-|-|-|-
> VeRA $^{r=12288}$ |0.042|20.65|9.00|23.5|38.7|85.3|54.3|50.4
> VeRA $^{r=1024}$ |0.015|17.80|1.15|23.6|35.5|82.1|53.3|48.6
>
> > **C3: Effect of weight-aware finetuning in cases of weak zero-shot performance of models**
>
> This is certainly a valid concern, especially when evaluating methods that reduce parameter count. Our experiments on the Math10k benchmark with LLaMA-1 (7B) address this issue (Table 2). Given LLaMA-1 7B’s low zero-shot accuracy of 11.0 on GSM8k [1], it can be considered weak for arithmetic reasoning which would require significant adaptation to perform well on Math10k. The results show:
>
> - Simply reducing parameters by lowering the rank in LoRA and DoRA leads to a significant performance drop. LoRA's accuracy with rank=16 is 50.9 vs. 48.9 with rank=2. With an equivalent parameter count (~0.05%), WAFT outperforms all baselines (LoRA, DoRA, VeRA, and VB-LoRA), and achieves accuracy close to LoRA $^{r=16}$. This demonstrates that generating the fine-tuning residuals from the pretrained weights improves adaptation with minimal parameters, whereas LoRA $^{r=2}$ struggles to achieve good performance.
>
> We have also included further experiments with the VTAB benchmark as suggested. We follow the same settings as the FGVC experiments (Section 4.5). The table below shows that WAFT can perform on par or better than baseline methods even on Specialized and Structured tasks on the VTAB benchmark.
>
> Method|Params (M)|Natural|Specialized|Structured|Avg
> -|-|-|-|-|-
> VPT|0.046|81.0|85.7|58.9|72.7
> BitFit|0.083|81.8|85.2|57.8|72.4
> LoRA |0.147|**82.0**|85.9|61.0|74.0
> WAFT|0.025 |**82.0**|**86.3**|**61.1**|**74.1**
>
>
> > **C4: More exploration could be done on the relevance of using the pre-trained weight for the matrix multiplication, for example is a properly scaled random matrix performing close to the pretrained weights?**
>
> We are not certain that we fully understand your concern. Were you suggesting to test a variant like this: converting WAFT update from $\Delta W^l= W^l\cdot \phi\cdot \psi$ to $\Delta W^l= \mathbb{W}^l\cdot \phi\cdot \psi$ in learning fine-tuning residual weights, where $\mathbb{W}^l$ is ``a properly scaled random matrix"?  It seems very unlikely that a random matrix can do better than the pretrained weights. With the random matrix, the overall fine-tuning residual weights $\Delta W^l$ would behave similar in spirit to VeRA. Could you kindly elaborate your concern?
>
> > **C5: Ablations on using other types of matrices for the multiplicative updates**
>
> The purpose of our simple formulation is that it enables memory and computationally efficient updates, since the low-rank structure and linear updates result in smaller matrix multiplications. Our ablations studies (Table 7) show that more complex updates are not necessary. Hence, we leave the further exploration to future work, and we would welcome any suggestions from the reviewers and the broader community.
>
> ---
> [1] Touvron et al., "LLaMA: Open and Efficient Foundation Language Models", ArXiv 2023

---

### Decision · Program_Chairs · 2025-05-01

**Decision:**

Accept (poster)

**Comment:**

This paper is focused on fine-tuning, seeking to improve on LoRA-style parameter efficient techniques. The authors introduce a simple multiplicative low-rank adapter (as opposed to the additive one in standard LoRA); the idea is related to other techniques sometimes used in representation editing.

The authors’ approach is very simple and works well in a range of settings. Reviewers were largely enthusiastic, with questions on how to extend the applicability of the work and on complexity. The authors cleared up all concerns. Solid paper overall.